# HNF1A recruits KDM6A to activate differentiated acinar cell programs that suppress pancreatic cancer

Mark Kalisz[1,2,3] (iD), Edgar Bernardo[4,5], Anthony Beucher[1], Miguel Angel Maestro[4,5], Natalia del Pozo[2,3], Irene Millán[2,3], Lena Haeberle[6] (iD), Martin Schlensog[6], Sami Alexander Safi[7], Wolfram Trudo Knoefel[7], Vanessa Grau[4,5], Matías de Vas[1], Karl B Shpargel[8], Eva Vaquero[9], Terry Magnuson[8], Sagrario Ortega[10], Irene Esposito[7], Francisco X Real[2,3,11] & Jorge Ferrer[1,4,5,*] (iD)

## Abstract

Defects in transcriptional regulators of pancreatic exocrine differentiation have been implicated in pancreatic tumorigenesis, but the molecular mechanisms are poorly understood. The locus encoding the transcription factor HNF1A harbors susceptibility variants for pancreatic ductal adenocarcinoma (PDAC), while *KDM6A*, encoding Lysine-specific demethylase 6A, carries somatic mutations in PDAC. Here, we show that pancreas-specific *Hnf1a* null mutant transcriptomes phenocopy those of *Kdm6a* mutations, and both defects synergize with *Kras*[G12D] to cause PDAC with sarcomatoid features. We combine genetic, epigenomic, and biochemical studies to show that HNF1A recruits KDM6A to genomic binding sites in pancreatic acinar cells. This remodels the acinar enhancer landscape, activates differentiated acinar cell programs, and indirectly suppresses oncogenic and epithelial–mesenchymal transition genes. We also identify a subset of non-classical PDAC samples that exhibit the *HNF1A/KDM6A*-deficient molecular phenotype. These findings provide direct genetic evidence that HNF1A deficiency promotes PDAC. They also connect the tumor-suppressive role of KDM6A deficiency with a cell-specific molecular mechanism that underlies PDAC subtype definition.

**Keywords** HNF1A; KDM6A; non-classical PDAC; pancreas; pancreas differentiation

**Subject Categories** Cancer; Chromatin, Transcription & Genomics

The EMBO Journal (2020) 39: e102808

See also: **S Bärthel et al.** (May 2020)

## Introduction

Pancreatic ductal adenocarcinoma (PDAC) is a leading cause of cancer mortality (Garrido-Laguna & Hidalgo, 2015). The incidence of PDAC is rising, yet current chemotherapies are generally ineffective (Ryan *et al*, 2014). Genomic analysis of PDAC has identified almost universal driver mutations in *KRAS*, *TP53*, *SMAD4,* and *CDKN2A*, among a long list of loci that show recurrent somatic mutations and structural variations (Jones *et al*, 2008; Biankin *et al*, 2012; Waddell *et al*, 2015; Bailey *et al*, 2016; Notta *et al*, 2016). A small subset of tumors is caused by germ-line mutations in DNA-repair genes (Waddell *et al*, 2015; Roberts *et al*, 2016), whereas GWAS have identified dozens of common variants that impact PDAC susceptibility (Childs *et al*, 2015; Klein *et al*, 2018). Genetic studies have therefore uncovered leads that promise to define molecular targets for future precision therapies.

Up to 18% of PDAC tumors carry mutations in *KDM6A* (Waddell *et al*, 2015), which encodes Lysine-specific demethylase 6A (KDM6A), a component of the MLL/COMPASS transcriptional co-regulatory complex (Cho *et al*, 2007). KDM6A catalyzes demethylation of histone H3K27me3, a modification associated with polycomb-mediated repression (Agger *et al*, 2007; Lan *et al*, 2007;

1 Section of Epigenomics and Disease, Department of Medicine, Imperial College London, London, UK
2 Epithelial Carcinogenesis Group, Spanish National Cancer Research Centre-CNIO, Madrid, Spain
3 CIBERONC, Madrid, Spain
4 Bioinformatics and Genomics Program, Centre for Genomic Regulation (CRG), The Barcelona Institute of Science and Technology (BIST), Barcelona, Spain
5 Centro de Investigación Biomédica en Red de Diabetes y Enfermedades Metabólicas Asociadas (CIBERDEM), Barcelona, Spain
6 Institute of Pathology, Heinrich-Heine University and University Hospital of Düsseldorf, Düsseldorf, Germany
7 Department of Surgery, Heinrich-Heine University and University Hospital of Düsseldorf, Düsseldorf, Germany
8 Department of Genetics and Lineberger Comprehensive Cancer Center, University of North Carolina at Chapel Hill, Chapel Hill, NC, USA
9 CiberEHD, Institut de Malalties Digestives i Metabòliques, Hospital Clínic, IDIBAPS, Barcelona, Spain
10 Transgenics Unit, Spanish National Cancer Research Centre-CNIO, Madrid, Spain
11 Departament de Ciències Experimentals i de la Salut, Universitat Pompeu Fabra, Barcelona, Spain
*Corresponding author. Tel: +34 933 160 197; E-mail: jorge.ferrer@crg.eu
[The copyright line for this article was changed on 16 April 2020 after original online publication.]

Lee *et al*, 2007). Most somatic pathogenic *KDM6A* mutations are likely to result in a loss of function, and mouse genetic studies have shown that *Kdm6a* and *Kras* mutations cooperate to promote PDAC (Mann *et al*, 2012; Andricovich *et al*, 2018). How KDM6A is recruited to its genomic targets in pancreatic cells, and the direct mechanisms through which it controls PDAC-relevant genetic programs are still poorly understood (Wang & Shilatifard, 2019).

There is increasing evidence that the transcriptional regulation of differentiated pancreatic exocrine cells is tightly linked to PDAC development and subtype definition (Stanger & Hebrok, 2013; Krah *et al*, 2015; Diaferia *et al*, 2016; Martinelli *et al*, 2016; Cobo *et al*, 2018). Little is known, however, about the underlying molecular underpinnings. We have examined HNF1A, a homeodomain transcriptional regulator of liver, gut, kidney, and pancreas, which has been proposed as a candidate pancreatic tumor suppressor (Molero *et al*, 2012; Stanger & Hebrok, 2013; Hoskins *et al*, 2014; Luo *et al*, 2015). Human heterozygous *HNF1A* loss-of-function mutations cause diabetes, in part because *HNF1A* promotes pancreatic β-cell proliferation, and mouse *Hnf1a* mutations prevent the formation of large T antigen-driven β-cell tumors (Servitja *et al*, 2009). The function of HNF1A, however, is cell-type specific (Servitja *et al*, 2009), and both co-expression network analysis of PDAC samples and *in vitro* studies suggest that *HNF1A* has a tumor-suppressive function in pancreatic exocrine cells (Hoskins *et al*, 2014; Luo *et al*, 2015). Furthermore, GWAS suggest that genetic variants in the *HNF1A* locus predispose to PDAC (Pierce & Ahsan, 2011; Klein *et al*, 2018). Despite these observations, there is currently no direct mouse or human genetic evidence to incriminate *HNF1A* deficiency in PDAC.

Here, we combine mouse genetics, transcriptomics, and genome binding studies to show that HNF1A is a major determinant for the recruitment of KDM6A to its genomic targets in acinar cells. This remodels the enhancer landscape of acinar cells and activates a broad epithelial cell transcriptional program that inhibits tumor suppressor pathways. We demonstrate that *Hnf1a* inactivation promotes *Kras*-induced PDAC, and partially phenocopies morphological features of *Kdm6a*-deficient PDAC. Finally, we define a subset of human tumors that exhibit *HNF1A/KDM6A*-deficient transcriptional programs. These findings, therefore, provide a molecular mechanism that connects the tumor-suppressive functions of KDM6A and pancreatic differentiation transcription factors.

# Results

## *Hnf1a* deficiency promotes Kras-induced oncogenesis

To directly test the role of *Hnf1a* in pancreatic carcinogenesis, we created a conditional *Hnf1a* loss-of-function allele ($Hnf1a^{LoxP}$) (Appendix Fig S1A) and used a $Pdx1^{Cre}$ transgene to delete *Hnf1a* in all pancreatic epithelial lineages (hereafter referred to as $Hnf1a^{pKO}$ mice, Appendix Fig S1B). HNF1A is normally expressed in pancreatic acinar and endocrine cells, but not in duct cells (Nammo *et al*, 2002), and $Hnf1a^{pKO}$ mice showed disrupted HNF1A expression in both acinar and endocrine cells (Appendix Fig S1C). As expected from previous studies of *Hnf1a* germ-line null mutants, this did not produce gross defects in pancreas organogenesis or tissue architecture (Appendix Fig S1D) although acinar cells displayed signs of

markedly increased proliferation (Pontoglio *et al*, 1996; Lee *et al*, 1998; Boj *et al*, 2001; Molero *et al*, 2012) (Fig 1A).

To determine whether *Hnf1a* interacts with *Kras*-induced carcinogenesis, we created mice with combined conditional *Hnf1a* and $Kras^{G12D}$ mutations, hereafter referred to as $Hnf1a^{pKO};Kras^{G12D}$ mice (Appendix Fig S1E). In the absence of *Hnf1a* mutant alleles, $Pdx1^{Cre}$-induced $Kras^{G12D}$ activation expectedly gave rise to occasional low-grade PanINs or acinar-to-ductal metaplasia (ADM) lesions by 2 months of age (Hingorani *et al*, 2003) (Fig 1B, E, H, K and N). $Hnf1a^{pKO};Kras^{G12D}$ mice showed no lesions at 7 days of age (Fig 1C and D), yet by weaning they had already developed focal ADM and desmoplastic reactions, which became more prominent as the mice aged (Fig 1F, G, I and J and data not shown). Eight-week-old $Hnf1a^{pKO};Kras^{G12D}$ mice additionally showed non-invasive atypical tubular complexes, higher-grade PanINs with luminal budding, desmoplastic reaction, and foci of spindle cell (mesenchymal) proliferation, some of which showed incipient infiltrative growth (Fig 1L, M, O and P). These findings indicate that pancreatic *Hnf1a* deficiency cooperates with *Kras* to promote sarcomatoid forms of PDAC.

## HNF1A activates an acinar differentiation program that inhibits oncogenic programs

To understand how *Hnf1a* deficiency promotes pancreatic cancer, we examined the transcriptional programs controlled by *Hnf1a* in pancreatic exocrine cells. Genetic lineage tracing studies in mice have shown that, despite the ductal morphology of PDAC, $Kras^{G12D}$-induced PDAC can originate from acinar cells that undergo ADM and PanIN, contrasting with intraductal papillary mucinous neoplasms that arise from duct cells (Kopp *et al*, 2012; von Figura *et al*, 2014). We therefore used a $Ptf1a^{Cre}$ allele, which ensured high-efficiency recombination of *Hnf1a* in mouse acinar cells, and more limited recombination in endocrine cells ($Ptf1a^{Cre}$; $Hnf1a^{LoxP/LoxP}$; hereafter referred to as $Hnf1a^{aKO}$) (Fig EV1A and B). Eight-week-old $Hnf1a^{aKO}$ mice were normoglycemic, and like $Hnf1a^{pKO}$ mice showed normal pancreatic histology (Fig EV1C). We profiled transcripts in pancreas from 8-week-old $Hnf1a^{aKO}$ mice and, despite the normal histology, found profound transcriptional changes (Fig 2A, Dataset EV1). We observed decreased expression of genes specific to differentiated acinar cells, including *Ptf1a*, *Pla2g1b*, *Serpini2,* and *Ctrb1* (Figs 2B and EV1D), and increased expression of genes specific to pancreatic mesenchymal cells (Fig 2B, Dataset EV2). Down-regulated genes were enriched in metabolic processes such as inositol phosphate turnover, amino acid metabolism, and protection against oxidative stress, whereas upregulated genes were enriched in annotations associated with the extracellular matrix (collagen formation, ECM-receptor interactions, integrin cell surface interactions) and complement activation (Fig 2C, Dataset EV2). We also observed activation of cell cycle-related pathways, cholesterol biosynthesis, and known oncogenic programs such as EMT, RAS, PI3K-AKT, STAT3, WNT, and MAPK signaling (Figs 2D and EV1E and F, Dataset EV2).

To assess which of these transcriptional changes reflects a direct function of HNF1A in acinar cells, we profiled genome-wide binding sites of HNF1A in adult pancreas (Fig EV1G), as well as H3K27 acetylation to mark active enhancers and promoters (Fig EV1H). HNF1A-bound genomic regions had canonical HNF1 recognition sequences in 405 out of the top 500 most significant binding sites

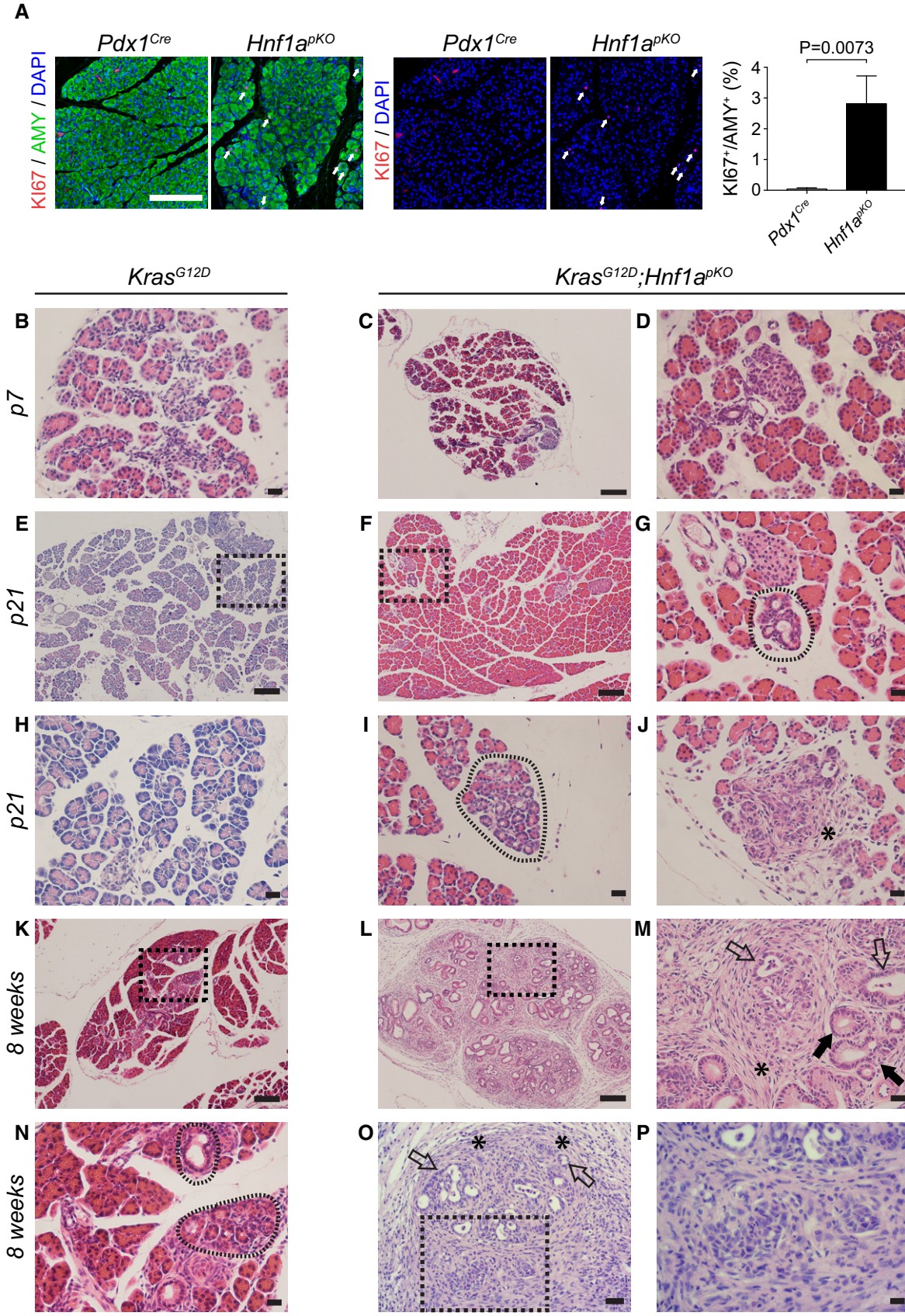

**Figure 1.**

Figure 1.  HNF1A deficiency leads to increased proliferation and promotes Kras-induced oncogenesis.

A    Representative immunofluorescence images and quantifications showing that 3-month-old $Hnf1a^{pKO}$ mice have increased number of KI67$^+$ (red) acinar cell nuclei co-staining with DAPI (blue) and Amylase (green). Arrows point to KI67$^+$ acinar cells in $Hnf1a^{pKO}$ mouse. Acinar proliferation is represented as the average of the KI67$^+$/Amylase$^+$ cell ratio. Quantifications were performed on 3 random fields from 3 $Pdx1^{Cre}$ and 3 $Hnf1a^{pKO}$ mice. P-values are from two-tailed Student's t-test. Representative H&E stainings of pancreata from $Kras^{G12D}$ and $Hnf1a^{pKO};Kras^{G12D}$ mice.

B–D    $Kras^{G12D}$ and $Hnf1a^{pKO};Kras^{G12D}$ mice have normal morphology at 7 days.

E–J    At 21 days, $Hnf1a^{pKO};Kras^{G12D}$ mice show acinar-to-ductal metaplasia (dashed encircled areas) and regions with desmoplastic reaction (asterisk), which are not observed in $Kras^{G12D}$ mice (E, H).

K–P    At 8 weeks, $Kras^{G12D}$ pancreas show occasional abnormal ductal structures (dashed encircled areas in N, which is a magnification of squared dotted box in K) and $Hnf1a^{pKO};Kras^{G12D}$ mice (L, M, O, P) present mucinous tubular complexes (black arrows), and more advanced PanINs with luminal budding (open arrows) including foci of spindle cell proliferation (asterisks) and incipient infiltrative growth (black dashed box area in O).

Data information: Black dashed boxes in (E, F, K, L and O) indicate magnified areas in (H, G, N, M and P) respectively. Scale bars indicate 200 μm (A), 100 μm (C, E, F, K, L), 50 μm (O), and 20 μm (B, D, G, H–J, M, N, P).

(81%), confirming the specificity of the assay (Fig EV1I), and they were expectedly enriched in enhancers and promoters (Fig EV1H). HNF1A binding was specifically enriched among genes that showed downregulation in $Hnf1a^{aKO}$ pancreas (odds ratio = 3.6 relative to all other genes, $P = 10^{-22}$, Fig 2E). This was consistent with HNF1A acting as an essential positive regulator of some of the genes to which it binds. HNF1A did not, however, show a higher number of binding near genes that were upregulated in $Hnf1a^{aKO}$ pancreas, compared with active genes that did not show altered regulation (odds ratio = 0.4, $P$ = N.S.; Figs 2E and EV1J). These studies, therefore, uncovered direct HNF1A-dependent genetic programs. They show that the function of HNF1A in pancreatic acinar cells entails direct transcriptional activation of a broad differentiated cell program that controls metabolic functions. They also revealed that HNF1A suppresses growth-promoting pathways and that, given the absence of enriched HNF1A binding to upregulated genes, this is largely mediated through indirect regulatory mechanisms.

## HNF1A-dependent programs in human acinar cells

We explored the relevance of mouse pancreatic exocrine HNF1A-dependent programs to human acinar cells. Given the lack of human acinar cell models to perform genetic manipulations, we examined transcriptomes from 328 human pancreas samples from the Genotype-Tissue Expression (GTEx) project (Consortium et al, 2017). We defined GTEx samples with highest and lowest deciles of $HNF1A$ mRNA expression ($n = 33$ per group) and asked if human orthologs of genes that were deregulated genes in $Hnf1a^{aKO}$ pancreas showed concordant changes. We observed that $Hnf1a^{aKO}$ up- and downregulated genes showed consistent up- and downregulation in $HNF1A$-low versus $HNF1A$-high human pancreas (median Log2 fold-difference [IQR]: upregulated genes 0.44 [0.05–0.81]; downregulated genes −0.16 [−0.38 to 0.05]; control genes 0.03 [−0.2 to 0.36]; Kruskal–Wallis $P < 10^{-4}$) (Fig 3A). Accordingly, 70 and 75% of $Hnf1a^{aKO}$ up- and downregulated genes showed differential expression in the same direction at nominal $P < 0.05$ in $HNF1A$-low versus $HNF1A$-high human pancreas, respectively. These observations suggested that HNF1A-dependent programs are largely conserved in mouse and human pancreatic cells.

## Deregulation of the HNF1A-dependent program in non-classical PDAC

We next studied HNF1A-dependent genes in human primary PDAC. We examined transcriptome data from the TCGA-PAAD study

(Cancer Genome Atlas Research Network. Electronic address & Cancer Genome Atlas Research, 2017) and found that human orthologs of deregulated genes in $Hnf1a^{aKO}$ pancreas showed concordant down- or upregulation in $non$-$classical$ tumor molecular subtypes—variously defined as quasimesenchymal, basal, and squamous-like PDAC (Collisson et al, 2011; Moffitt et al, 2015; Bailey et al, 2016) (Fig 3B). We employed these same $Hnf1a^{aKO}$ deregulated gene sets to cluster human tumor samples from TCGA-PAAD and ICGC-PACA cohorts (Bailey et al, 2016) using non-negative matrix factorization. This exposed a cluster of predominantly non-classical tumors that showed transcriptional changes in the same direction as $Hnf1a$-deficient pancreas (HNF1A cluster 3, Fig EV2A).

We next sought to identify tumors with most pronounced HNF1A-deficient function. Because our findings showed that HNF1A primarily functions as a direct activator, we defined tumors with most pronounced downregulation of genes that were both HNF1A-bound and downregulated in $Hnf1a$ mutant mice (hereafter referred to as HNF1A loss of function, or HNF1A LoF tumors). We found that HNF1A LoF tumors showed a remarkable concordance with tumors that were consistently classified as non-classical molecular subtypes in independent studies, and were therefore significantly enriched in quasimesenchymal (Fisher's $P = 1.7 \times 10^{-14}$), basal ($P = 2.9 \times 10^{-6}$), and squamous-like ($P = 1.1 \times 10^{-9}$) subtypes (Fig 3C). They also had increased $TP63$ mRNA, a marker of squamous-like PDAC molecular subtype (Bailey et al, 2016; Andricovich et al, 2018; Somerville et al, 2018) (Fig EV2B).

HNF1A LoF tumors—as well as non-classical PDAC—showed significantly lower $HNF1A$ mRNA levels (Kruskal–Wallis $P = 0.0064$), although there was considerable overlap with control tumors, suggesting that abnormal HNF1A function cannot be exclusively explained by differences in $HNF1A$ expression (Figs 3D and EV2C). Furthermore, although tumors with an HNF1A LoF signature also had non-classical molecular signatures—which correlate with high histological grade—HNF1A immunoreactivity was not significantly lower in high histological grade (poorly differentiated) tumors in tissue microarrays of human PDAC ($n = 102$) (Fig EV2E). These results, therefore, revealed that a subset of non-classical PDAC tumors had a transcriptional signature that was consistent with abnormal HNF1A function.

## HNF1A LoF signature in KDM6A-deficient non-classical PDAC

Although a subset of non-classical PDAC samples showed a gene expression profile that resembled that of pancreatic $Hnf1a$-mutant mice, $HNF1A$ mRNA levels were not invariably altered, and so far,

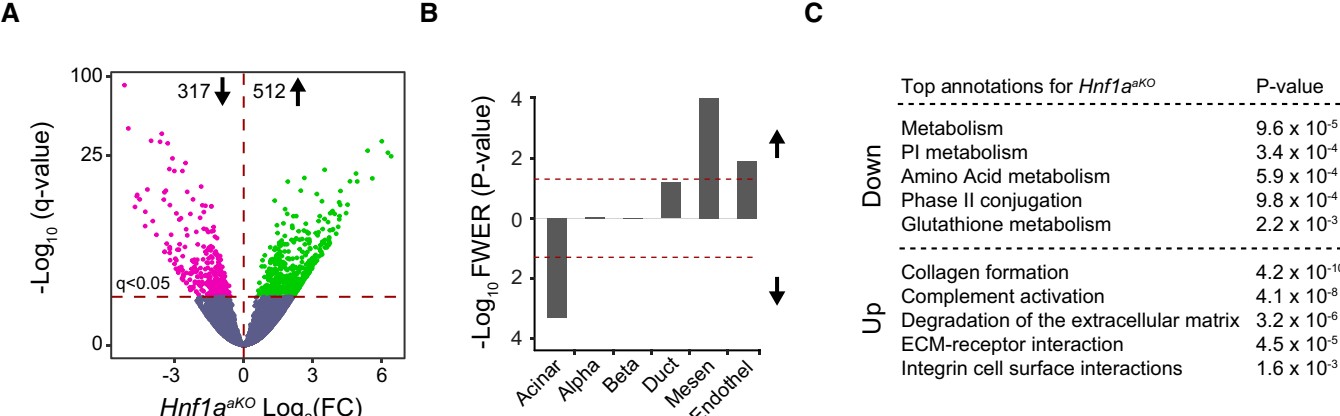

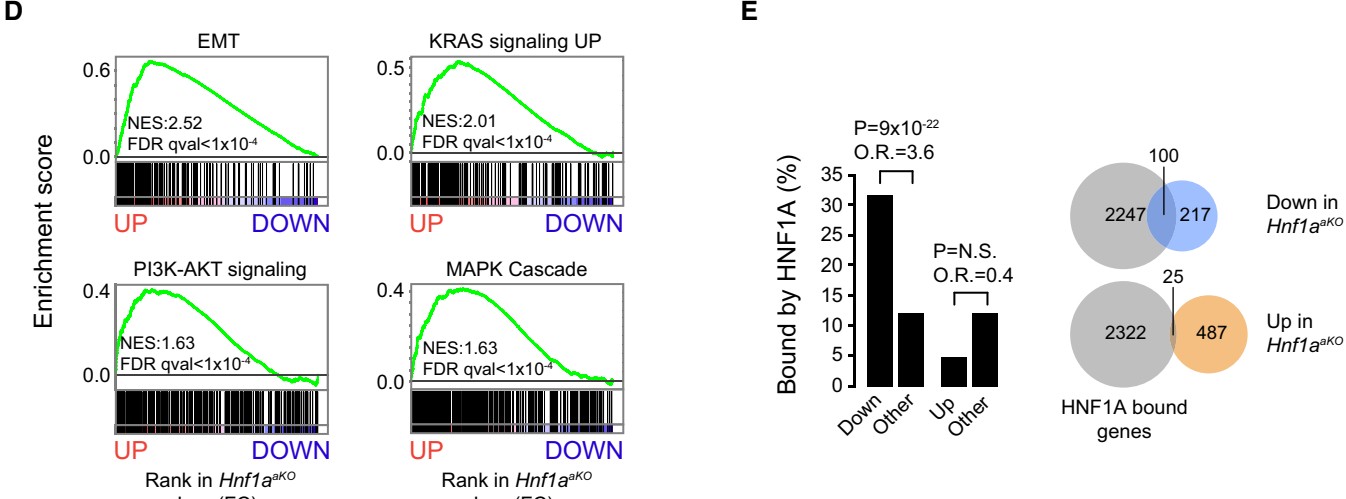

**Figure 2. HNF1A regulates gene programs essential to acinar cell identity, metabolism, and proliferation.**

A   Fold change (FC) in transcripts in *Hnf1a^aKO* versus control pancreas, plotted against significance ($-\text{Log}_{10}$ q; genes significant at $q < 0.05$ are shown as colored dots above the horizontal line).

B   GSEA showing that genes specific to differentiated acinar cells were downregulated in *Hnf1a^aKO* pancreas, but not genes specific to islets or duct cells. Upregulated genes were enriched in genes specific to mesenchymal cells. Lineage-enriched genes were obtained from Muraro *et al* (2016).

C   Top functional annotations for differentially expressed genes in *Hnf1a^aKO* pancreas.

D   GSEA revealed that *Hnf1a^aKO* pancreas showed increased transcripts involved in oncogenic pathways such as EMT, MAPK, KRAS, PI3K-AKT.

E   HNF1A promotes transcriptional activation of direct target genes. Left: HNF1A-bound genes were enriched among genes that showed downregulation in *Hnf1a* mutants, but not among upregulated genes. *P*-values and odds ratios (O.R.) calculated by Fisher's exact test. Right: Venn diagrams showing overlap of HNF1A-bound genes with genes that were downregulated and upregulated in *Hnf1a* mutant pancreas.

recurrent somatic *HNF1A* mutations have not been reported in PDAC. This raised the question of why non-classical PDAC shows abnormal HNF1A function. Non-classical (e.g., squamous-like) PDAC have been shown to express low *KDM6A* mRNA and are enriched in *KDM6A* somatic genomic defects (Bailey *et al*, 2016; Andricovich *et al*, 2018). Our studies also showed decreased *KDM6A* mRNA in non-classical PDAC ($P < 0.001$) (Fig EV2D) and decreased KDM6A immunoreactivity in poorly differentiated tumors ($P = 0.03$, $n = 94$) (Fig EV2F). Importantly, tumors with *HNF1A* LoF phenotypes showed decreased *KDM6A* mRNA (median [IQR]:

41.9 [32.2–50.5] in LoF tumors, versus 57.3 [44.3–64.3] and 79.4 [53.1–104.2] in control tumors, Kruskal–Wallis $P < 0.0001$) (Fig 3E). Furthermore, analysis of the Australian ICGC-PACA data revealed putative loss-of-function *KDM6A* mutations in 19% of tumors showing *HNF1A* LoF phenotypes, compared with 2% of all other tumors (Fisher's $P = 0.005$) (Fig 3C). Likewise, tumors with *KDM6A* putative loss-of-function mutations showed abnormal *HNF1A*-dependent programs (Fig 3C). Collectively, these correlations hinted at a mechanistic link between *KDM6A*- and *HNF1A*-deficient phenotypes in non-classical PDAC.

## Kdm6a-dependent Kras-induced oncogenesis

To examine the relationship between *Hnf1a* and *Kdm6a* deficiency in pancreatic cancer, we generated mice with pancreas-specific

inactivating *Kdm6a* mutations and oncogenic *Kras* mutations [*Pdx1^Cre^*, *Kdm6a^LoxP/LoxP^*, *Kras^G12D^*, hereafter referred to as *Kdm6a^pKO^;Kras^G12D^* mice (Appendix Fig S2A and B)]. KDM6A is normally expressed in all pancreatic cell types and was efficiently

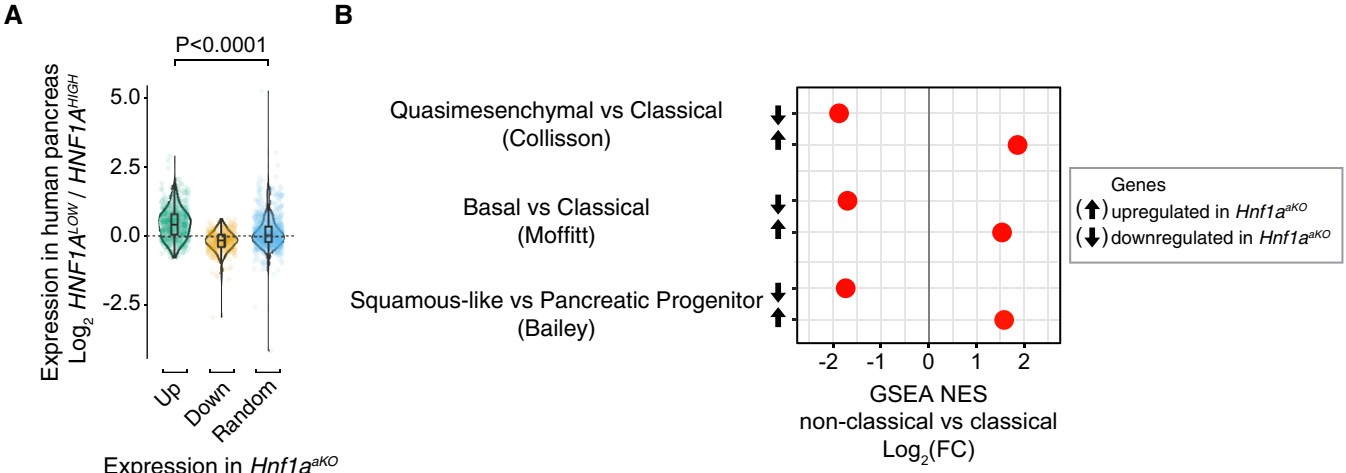

Figure 3.

**Figure 3. Human non-classical and *KDM6A*-deficient PDAC exhibit *HNF1A*-deficient phenotypes.**

A   Human orthologs of genes that were up- and downregulated in *Hnf1a^aKO* pancreas were also up- and downregulated in human pancreas with low versus high *HNF1A* expression (lowest versus highest expression deciles, respectively). A random list of 717 genes controlled for similar expression levels was used for comparison. Violin plots include median and interquartile ranges. Dots are average values for each gene. Kruskal–Wallis $P < 0.0001$.

B   GSEA demonstrates that down- or upregulated genes in *Hnf1a^aKO* mice (downward or upward arrows) showed down- or upregulation, respectively, in gene lists ranked by differential expression in non-classical versus classical PDAC molecular subtypes from the TCGA-PAAD study (Cancer Genome Atlas Research Network, Electronic Address Aadhe, Cancer Genome Atlas Research N, 2017). All enrichments had GSEA FDR *q*-values $< 0.01$.

C   Analysis of HNF1A function in 121 high-purity cases of the ICGC-PACA-AU cohort identified tumors with most pronounced downregulation of direct HNF1A target genes. We performed GSEA with a gene set of 106 human orthologs of HNF1A direct targets showing downregulation in *Hnf1a^aKO* pancreas. For each tumor sample, we performed differential expression against all other samples and used GSEA to ascertain abnormal expression of the mouse HNF1A-dependent gene set in the tumor. Samples were ranked by the resulting normalized enrichment score (NES) and classified as either *HNF1A* LoF samples (purple, NES $< 0$; $P < 0.05$), or *Control 1* (beige, NES $< 0$; $P > 0.05$) and *Control 2* (gray, NES $> 0$). *HNF1A* LoF samples were predominantly non-classical tumors (Collisson *et al*, 2011; Moffitt *et al*, 2015; Bailey *et al*, 2016). Putative loss-of-function *KDM6A* mutations (*KDM6A* LoF mutants) were found in 19% of *HNF1A* LoF tumors versus 2% of all others (Fisher's $P = 0.005$). *KDM6A* mutations were considered functional if classified as "high" functional impact in ICGC (small $\leq$ 200-bp deletions/insertions, single base substitutions), or as likely loss-of-function structural variants in Bailey *et al* (2016), all of which were frame-shift mutations. Other *KDM6A* mutations were classified as unknown. Heatmaps show *Z*-score-normalized expression of deregulated genes in *Hnf1a^aKO* pancreas. We confirmed that 85% of 106 downregulated and 60% of genes of 146 upregulated human orthologs showed differential expression across the 3 *HNF1A* profiles ($q < 0.05$, SAM multiclass analysis).

D   *HNF1A* mRNA levels differed in *HNF1A* LoF and control groups (Kruskal–Wallis, $P < 0.01$), despite considerable variability and overlap between groups.

E   *KDM6A* mRNA levels were downregulated in *HNF1A* LoF tumors (Kruskal–Wallis, $P < 0.001$).

Data information: Box plots in (D and E) show *HNF1A* and *KDM6A* expression in *HNF1A* LoF tumors ($n = 26$) and *Control 1* ($n = 39$) and *Control 2* ($n = 57$) tumors. The horizontal central line marks the median. Box limits indicate the first and third quartiles, and whiskers extend to highest and lowest data points within 1.5× interquartile range outside box limits.

excised in *Kdm6a^pKO* pancreas (Appendix Fig S2C and D). We focused on female mice because KDM6A is encoded in the X chromosome, and males harbor a Y chromosome paralog named *Uty*. *Kdm6a^pKO;Kras^G12D* mice showed normal pancreas morphology at 7 days of age (Fig 4A–C), but subsequently rapidly developed invasive PDAC. Early tumors were apparent by weaning, showing prominent signs of spindle cell proliferation, sarcomatoid morphology with occasional glandular tumor components, regions with widespread signs of ADM and abundant desmoplastic reaction (Fig 4E, F, H and I). Subsequently, tumors showed massive remodeling and very rapid infiltrative growth (Fig 4K, L, N and O), leading to the death of all *Kdm6a^pKO;Kras^G12D* female mice by 15–16 weeks of age (Appendix Fig S2E). This contrasted with control *Kras^G12D* mice, which showed normal morphology at weaning (Fig 4D and G) and expectedly only presented occasional acinar-to-ductal metaplasia at 8 weeks (Fig 4J and M). These findings were generally consistent with recently reported observations in 6-week-old *Kdm6a* mutant mice (Andricovich *et al*, 2018), with the exception that we have not observed histological signs of squamous differentiation. Male mice presented delayed mortality compared to females, suggesting partial *Uty* compensatory tumor suppressor functions, as reported previously (Andricovich *et al*, 2018; Gozdecka *et al*, 2018). These findings, therefore, showed *Kdm6a*-dependent sarcomatoid PDAC lesions that are reminiscent of those observed in *Hnf1a^pKO;Kras^G12D* mice, although *Kdm6a*-deficient tumors showed earlier age of onset and much more rapid growth. These results were consistent

with the overlapping *KDM6A*- and *HNF1A*-deficient molecular signatures of human tumors.

**KDM6A activates an acinar cell growth-suppressing program**

The analysis of molecular signatures in human tumors, and the semblance of mouse genetic phenotypes, suggested that KDM6A and HNF1A might control common tumor-suppressive programs in pancreatic exocrine cells. We thus studied the transcriptional function of KDM6A in non-tumoral, differentiated pancreatic exocrine cells, following a strategy similar to that described above for HNF1A. We studied female *Kdm6a^pKO* mice, in which the *Pdx1^Cre* transgene led to pancreatic inactivation of *Kdm6a* in most cells from all pancreatic epithelial lineages as early as e15.5, and a marked reduction of KDM6A protein at weaning (Fig EV3A, Appendix Fig S2D). *Kdm6a^pKO* mice were born at the expected Mendelian ratio and appeared healthy. They had normal glycemia at 12 weeks of age (Fig EV3B) and unaltered pancreas morphology at weaning, indicating that, like HNF1A, KDM6A is dispensable for pancreas organogenesis (Fig EV3C–F). In analogy to observations in *Hnf1a^aKO* mice, acinar cells showed increased proliferation at weaning (Fig EV3I), although by 8 weeks of age some mice showed acinar cell attrition and atrophic pancreatic lobules (Fig EV3G and H).

To assess the transcriptional function of KDM6A in the pancreas, we analyzed pancreatic RNA (> 98% of which originates from

**Figure 4. *Kdm6a*-dependent Kras-induced oncogenesis.**

H&E staining of pancreata from *Kdm6a^pKO;Kras^G12D* and *Kras^G12D* mice.

A–C   Seven-day-old *Kras^G12D* and *Kdm6a^pKO;Kras^G12D* mice showed normal morphology.

D–I   At 21 days, *Kras^G12D* mice showed normal pancreas morphology (dotted area in D is shown at large magnification in G), whereas *Kdm6a^pKO;Kras^G12D* mice already show acinar-to-ductal metaplasia (dashed encircled area in F), spindle cell proliferation (asterisks in E and H), sarcomatoid architecture (I), and desmoplastic reaction (black arrowhead in H).

J–O   *Kras^G12D* mice present occasional acinar-to-ductal metaplasia and low-grade PanINs at 8 weeks (M is a magnification of squared dotted box in J and see Fig 1N), whereas at the same age pancreas from *Kdm6a^pKO;Kras^G12D* mice show massive remodeling (K), extensive acinar-to-ductal metaplasia (L and dashed encircled area in N), cancer with prominent spindle/mesenchymal proliferation, infiltrative growth (black arrows and asterisk, respectively, in O) and abundant stroma (black arrowheads in N).

Data information: Scale bars: 100 μm (B, D, E, J, K) or 20 μm (A, C, F–I, L–N).

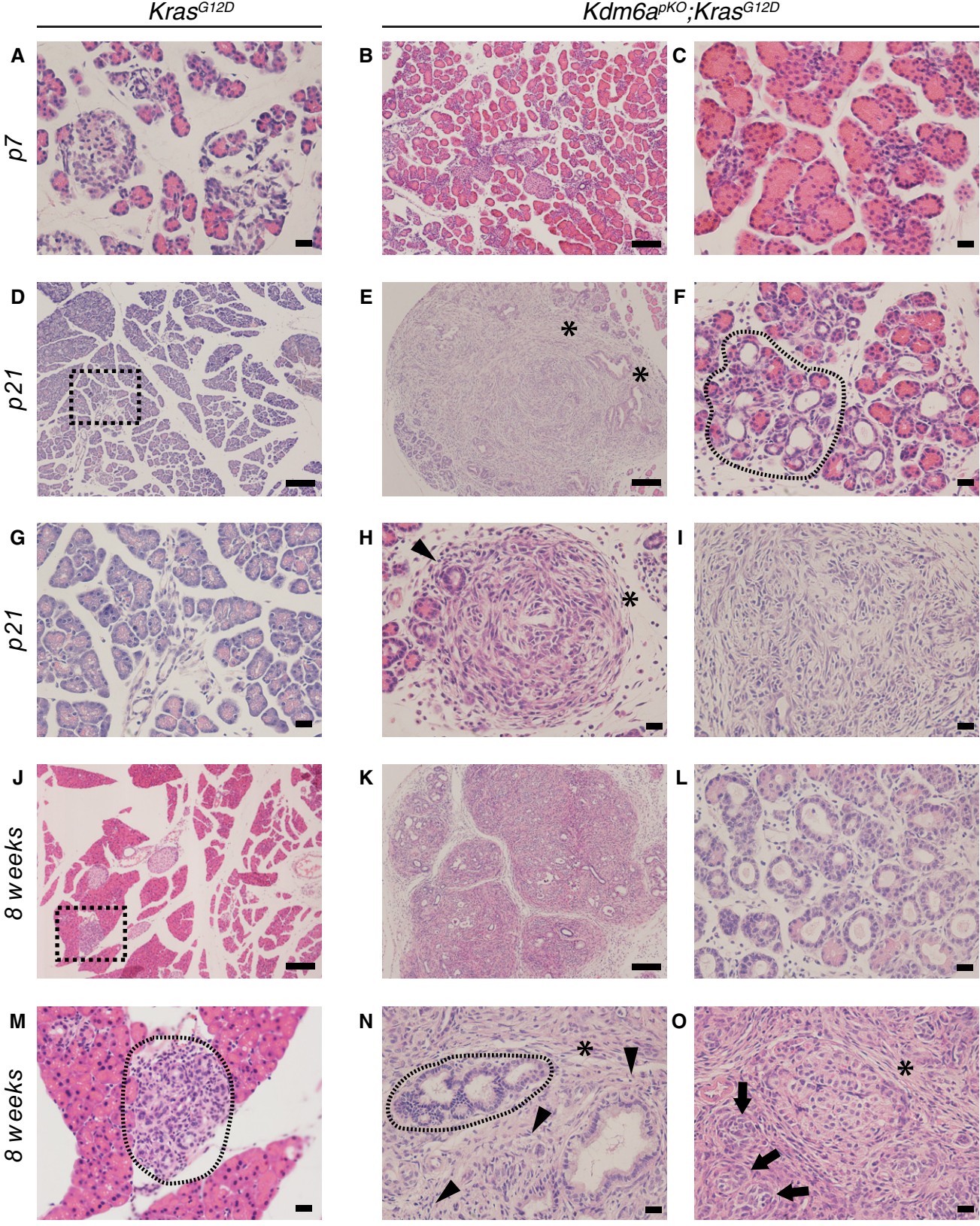

**Figure 4.**

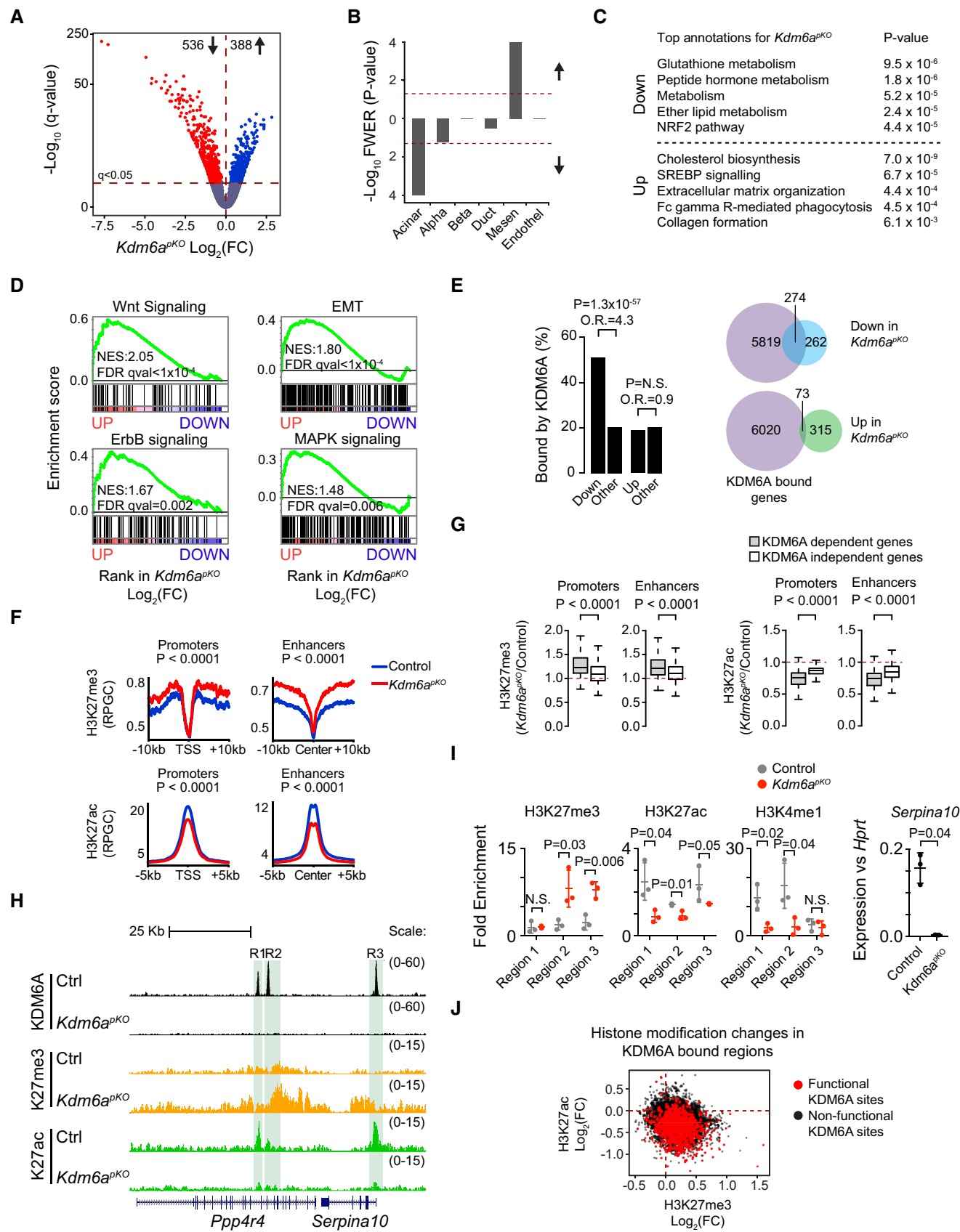

**Figure 5.**

Figure 5.  KDM6A promotes acinar cell differentiation gene programs and suppresses oncogenic pathways.

A  Fold change (FC) and significance for transcripts in *Kdm6a*^*pKO*^ versus control pancreas. Genes significant at *q* < 0.05 are shown as colored dots above the horizontal line.

B  GSEA showing that genes specific to differentiated acinar cells were downregulated in *KDM6A-deficient* pancreas, but not genes specific to islets or duct cells. Upregulated genes were enriched in genes specific to mesenchymal cells. Lineage-enriched genes were obtained from Muraro *et al* (2016).

C  Most enriched functional annotations in genes differentially expressed in *Kdm6a*^*pKO*^.

D  GSEA showed increased expression of indicated oncogenic pathway genes in *Kdm6a*^*pKO*^ pancreas.

E  KDM6A functions as a transcriptional activator of direct target genes in pancreatic cells. Left: Downregulated, but not upregulated, genes in *Kdm6a*^*pKO*^ pancreas were enriched for KDM6A binding. *P*-values and odds ratios (O.R.) were calculated by Fisher's exact test. Right: Venn diagrams showing overlap of down- and upregulated genes with KDM6A binding.

F  KDM6A-bound enhancers and promoters showed increased H3K27me3 and decreased H3K27ac in *Kdm6a*^*pKO*^ pancreas.

G  Genes that were downregulated in *Kdm6a*^*pKO*^ pancreas (KDM6A-dependent genes) showed greatest changes in histone marks. Box plots show fold-changes of H3K27me3 (left) and H3K27ac (right) signals in *Kdm6a*^*pKO*^ compared to control. Signals were analyzed in KDM6A-bound regions in promoters and enhancers of genes that were KDM6A-dependent (gray; *n* = 420 regions) and KDM6A-independent (white; *n* = 8,035 regions). The signals are average values from ChIP-seq experiments in two biological replicates. The horizontal central line marks the median. Box limits indicate the first and third quartiles, and whiskers extend to highest and lowest data points within 1.5× IQR outside box limits. *P*-values were determined by two-tailed Mann–Whitney *U*-test.

H  Changes in KDM6A, H3K27me3, and H3K27ac ChIP-seq profiles in the *Ppp4r4-Serpina10* locus in *Kdm6a*^*pKO*^ pancreas.

I  qPCR of H3K27me3, H3K27ac, and H3K4me1 changes in promoter and enhancer regions highlighted in (H) (R1, R2, R3), and *Serpin10* mRNA. Error bars show ± SD, and *P*-values are from two-tailed Student's *t*-test, *n* = 3.

J  KDM6A-bound genes that showed decreased expression in *Kdm6a*^*pKO*^ pancreas (*functional* KDM6A binding sites) showed simultaneous gain of H3K27me3 and loss and of H3K27ac.

exocrine cells) from 4-day-old female *Kdm6a*^*pKO*^ mice. This revealed profound transcriptional changes, with downregulation of acinar-specific genes (but not duct or islet genes) and upregulation of mesenchymal genes (Fig 5A and B, Dataset EV3). Downregulated genes were associated with metabolic pathways including glutathione, ether lipid, and NRF2 (antioxidant) metabolism (Fig 5C). Upregulated genes in *Kdm6a*^*pKO*^ mice were enriched in programs regulating cholesterol biosynthesis, extracellular matrix organization, and the innate immune response (Fc gamma R-mediated phagocytosis) (Fig 5C). GSEA showed prominent enrichment in additional annotations related to oncogenic pathways, including oncostatin M, NFKB, Wnt, EMT, ErbB, and MAPK signaling (Figs 5D and EV3J and K, Dataset EV4). The biological pathways that were down- and upregulated in *Kdm6a*^*pKO*^ mice were generally deregulated in the same direction as in *Hnf1a*^*aKO*^ (Fig EV3L, Dataset EV5). This suggested that *Kdm6a* could exert its tumor suppressor function through broadly similar pathways as *Hnf1a*, namely through the maintenance of acinar differentiated cell programs and inhibition of growth-promoting pathways.

### Shared transcriptomes in *Kdm6a*^*pKO*^ mice and *KDM6A*-mutant human PDAC

The existence of *KDM6A* mutations in some human tumors allowed us to directly assess the relevance of the transcriptional phenotype of *Kdm6a*^*pKO*^ mice to human PDAC. GSEA showed that up- and downregulated genes from tumors with *KDM6A* putative loss-of-function mutations were concordantly up- and downregulated in *Kdm6a*^*pKO*^ pancreas (Fig EV3M). Furthermore, *KDM6A* mRNA was decreased in human tumors that showed greatest deregulation of genes downregulated in *Kdm6a*^*pKO*^ pancreas (Fig EV3N). Likewise, non-classical PDAC tumors, which exhibit decreased *KDM6A* expression, showed down- and upregulation of deregulated genes in *KDM6A*^*pKO*^ pancreas (FDR *q* < 0.0001 and 0.04, respectively; Fig EV3O). The transcriptional changes observed in *Kdm6a*^*pKO*^ mice are, therefore, directly relevant to a subset of human PDAC tumors.

### KDM6A co-activates acinar differentiated cell programs

To further understand the direct molecular mechanisms whereby KDM6A regulates genetic programs relevant to PDAC, and the potential link to HNF1A, we profiled KDM6A-bound genomic sites in pancreas from 4-day-old mice, and integrated these data with transcriptome changes in *Kdm6a*^*pKO*^ pancreas. We identified 8455 KDM6A binding sites (Fig EV3P) and confirmed their specificity by showing the absence of binding in *Kdm6a*^*pKO*^ pancreas (Fig EV3Q and R). The vast majority of KDM6A binding sites were located in active enhancers and promoters (Fig EV3S), and they were enriched in genes that were downregulated in *Kdm6a*^*pKO*^ mice, but not in those that showed upregulation (Figs 5E and EV3Q and R, and EV1J). KDM6A was, consequently, preferentially bound near acinar-enriched genes, but not to genes involved in EMT or other pathways that showed induction in *Kdm6a*^*pKO*^ pancreas. Thus, although KDM6A has been reported to have transcriptional activating and repressive functions (Gozdecka *et al*, 2018), its predominant direct function in pancreatic cells appears to be the transcriptional activation of gene targets, and indirect inhibition of many other genes.

KDM6A is a H3K27 demethylase, although previous studies have also demonstrated KDM6A functions that are independent from this catalytic activity (Miller *et al*, 2010; Shpargel *et al*, 2012, 2017; Vandamme *et al*, 2012; Andricovich *et al*, 2018; Gozdecka *et al*, 2018). We observed that in pancreas from *Kdm6a*^*pKO*^ mice KDM6A-bound enhancers and promoters showed increased H3K27me3 (Fig 5F), particularly at genes that were downregulated in *Kdm6a*^*pKO*^ mice (Figs 5G–I and EV3Q and R). Consistent with KDM6A forming part of complexes that contain histone acetyl transferases P300/CBP and histone methyltransferases MLL1/2 (Tie *et al*, 2012; Zha *et al*, 2015; Wang *et al*, 2017), *Kdm6a* mutants also showed decreased H3K27ac and H3K4me1 in KDM6A-bound regions (Figs 5F–I and EV3Q and R). KDM6A-bound sites, therefore, often showed concomitant H3K27me3 gain and H3K27ac loss in mutant cells (Fig 5J). Collectively, these results indicate that KDM6A has a profound direct influence on chromatin states and transcriptional activity in pancreatic cells.

## HNF1A and KDM6A share functional targets in acinar cells

KDM6A is not a sequence-specific DNA binding protein, and the mechanisms that recruit KDM6A to its genomic sites are poorly understood. We therefore examined KDM6A-bound sequences to identify candidate transcription factors (TFs) that recruit KDM6A in pancreatic cells, and specifically focused on KDM6A-bound sites associated with transcriptional changes in *Kdm6a^pKO* mice.

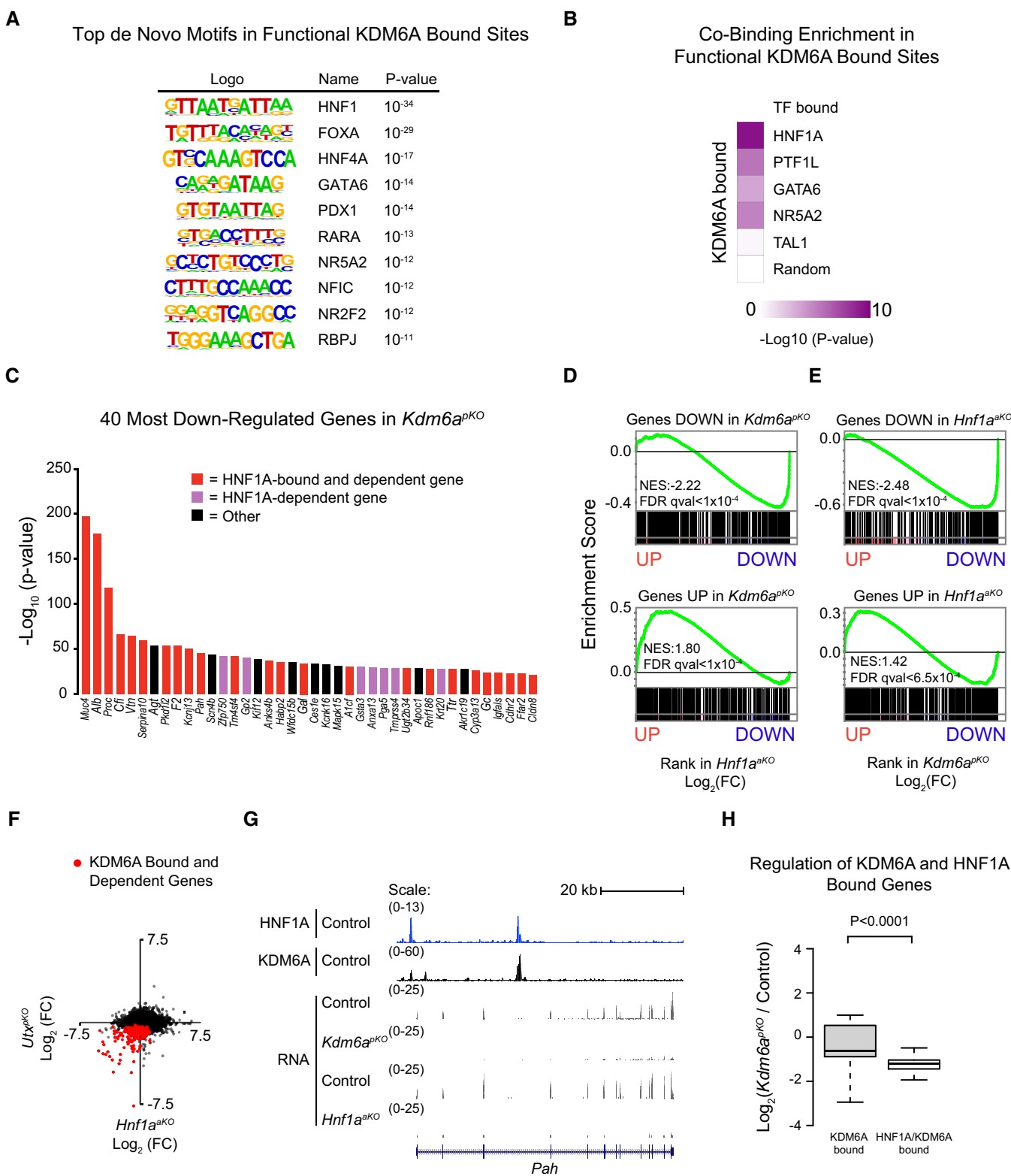

**Figure 6.**

**Figure 6. KDM6A and HNF1A regulate shared target genes.**

A  Motif analysis in functional KDM6A-bound regions, showing top ten *de novo* motifs ranked by *P*-value determined by HOMER software.
B  Co-binding analysis in functional KDM6A-bound enhancer and promoter regions revealed that HNF1A was the most enriched co-bound TF among three other acinar cell TFs. Binding regions of TAL1 in a non-pancreatic cell type and random binding sites were used as negative controls. *P*-values were determined by Fisher's exact test for peak comparisons using all enhancer and promoter regions as background.
C  The most downregulated genes in *Kdm6a^pKO* pancreas are shown ranked by q-value and are almost invariably bound by HNF1A and downregulated in *Hnf1a^aKO* pancreas, or known to be direct HNF1A-dependent target genes from other studies (red and purple, respectively).
D, E  GSEA analysis on the *Hnf1a^aKO* and *Kdm6a^pKO* ranked-ordered gene lists versus their reciprocal up- or downregulated gene sets, demonstrated that KDM6A and HNF1A regulate similar genes.
F  Expression changes in *Hnf1a^aKO* and *Kdm6a^pKO* pancreas, showing that genes bound by KDM6A and downregulated in *Kdm6a^pKO* pancreas (red dots) were generally downregulated in *Hnf1a^aKO* pancreas.
G  HNF1A and KDM6A co-occupy the same regions in *Pah*, which is downregulated in *Hnf1a* and *Kdm6a* knock-out pancreas.
H  Genes that were co-bound by KDM6A and HNF1A showed greatest downregulation in *Kdm6a^pKO* pancreas, compared with KDM6A-bound genes that were not bound by HNF1A. Box plots show median and IQR of Log$_2$ TPM fold-changes and whiskers extend to highest and lowest data points within 1.5× IQR outside box limits. *P*-values were determined by two-tailed Student's *t*-tests and $n = 4$ replicates per condition.

These *functional* KDM6A-bound regions were enriched in *in silico*-predicted DNA recognition sequences for canonical acinar TFs such as FOXA, GATA6, NR5A2, and RBPJL, although the most enriched sequence was the HNF1 recognition motif (Fig 6A). Consistently, functional KDM6A-bound regions were enriched for *in vivo* binding sites for multiple acinar cell TFs, yet showed the highest enrichment for HNF1A binding (Fig 6B). These findings, therefore, pointed to a strong overlap between KDM6A and HNF1A occupancy in pancreatic cells.

Consistent with KDM6A and HNF1A co-occupancy, 31 of the 40 most downregulated genes in *Kdm6a^pKO* pancreas were also downregulated in *Hnf1a^aKO* pancreas or are known to be HNF1A-dependent target genes (Fig 6C). Accordingly, gene set enrichment analysis (GSEA) showed that genes that were significantly down- or upregulated in *Kdm6a^pKO* mice were preferentially down- or upregulated in *Hnf1a^aKO* mice (Fig 6D and E for the converse comparison). We verified downregulation of selected HNF1A/KDM6A co-occupied genes by quantitative PCR in *Hnf1a* knock-out pancreas (Fig EV4A–D) and in *Kdm6a* knock-out clonal cell lines (Fig EV5B–F).

We found that KDM6A-bound genes that showed downregulation in *Kdm6a^pKO* mice were largely downregulated in the pancreas of *Hnf1a^aKO* (Fig 6F and G). Importantly, KDM6A-bound genes showed downregulation in *Kdm6a^pKO* if they were co-bound by HNF1A, but showed more marginal changes when they were not co-bound by HNF1A, suggesting that HNF1A is critical for KDM6A function in the pancreas (Fig 6H).

Among genes that were co-bound by HNF1A/KDM6A and downregulated in *Kdm6a^pKO* and *Hnf1a^aKO* pancreas, we identified several negative regulators of EMT such as *Foxa3* (Jagle *et al*, 2017) and *Deptor* (Chen *et al*, 2019) as well as negative MAPK regulators such as *Gstp1* (Ruscoe *et al*, 2001; Xue *et al*, 2005) and *Ptprj* (Sacco *et al*, 2009) (Fig EV3T and U). These findings, therefore, indicated that the derangement of shared biological pathways in *Kdm6a^pKO* and *Hnf1a^aKO* mice (shown in Fig EV3L) was largely due to the deregulation of common direct target genes.

We further found that transcriptional changes in HNF1A- and KDM6A-deficient pancreas were also recapitulated in *Kdm6a*-mutant *Kras^G12D* tumors (Andricovich *et al*, 2018) (Fig EV5G and H). This provided further evidence that HNF1A function is not only relevant to *Kdm6a*-regulated programs in the non-tumoral pancreas, but also to *Kdm6a*-deficient cancer.

Collectively, these findings indicate that HNF1A and KDM6A target common genomic sites and regulate shared genetic programs in pancreatic acinar cells.

### HNF1A recruits KDM6A to genomic targets in pancreatic cells

To define the molecular mechanisms that link KDM6A and HNF1A function, we performed co-immunoprecipitation experiments using purified nuclei from a mouse acinar cell line. This showed that HNF1A and KDM6A form part of a common complex in pancreatic cells (Fig 7A). To test whether this interaction could mediate the recruitment of KDM6A by HNF1A to its genomic targets, we performed ChIP-seq for KDM6A using chromatin from pancreas of wild-type and *Hnf1a^−/−* mice, which had unchanged KDM6A levels (Fig 7B). This showed that KDM6A binding to genomic targets was drastically reduced in *Hnf1a^−/−* pancreas (Fig 7C). Importantly, regions with reduced KDM6A binding in *Hnf1a^−/−* pancreas were bound by HNF1A and carried HNF1 motifs (Figs 7D–F and EV4A–D). These findings, therefore, showed that HNF1A recruits KDM6A to genomic targets. Consistent with the functional importance of this recruitment, most genes associated with reduced KDM6A binding in *Hnf1a*-deficient pancreas were direct HNF1A target genes with decreased expression in *Hnf1a*-deficient pancreas (Figs 7G and EV4A–D). By contrast, HNF1A binding was not affected in *Kdm6a^pKO* pancreas (Fig EV5A) or in co-bound genes in *Kdm6a* knock-out cells, indicating that, despite the functional interdependence of both proteins, KDM6A is not required for HNF1A binding to chromatin (Fig EV5B–F).

These results, therefore, show that HNF1A and KDM6A interact in a common complex, and this enables the recruitment of KDM6A to its functional targets in pancreatic cells. Collectively, these findings suggest a mechanistic model (Fig 7H) that explains the overlapping genomic phenotypes of HNF1A and KDM6A deficiency in genetic mouse models and human tumors, and their role in the determination of PDAC subtypes.

## Discussion

In this study, we provide a direct genetic demonstration that *HNF1A*, which encodes for a homeodomain transcription factor best known for its causal role in autosomal dominant diabetes

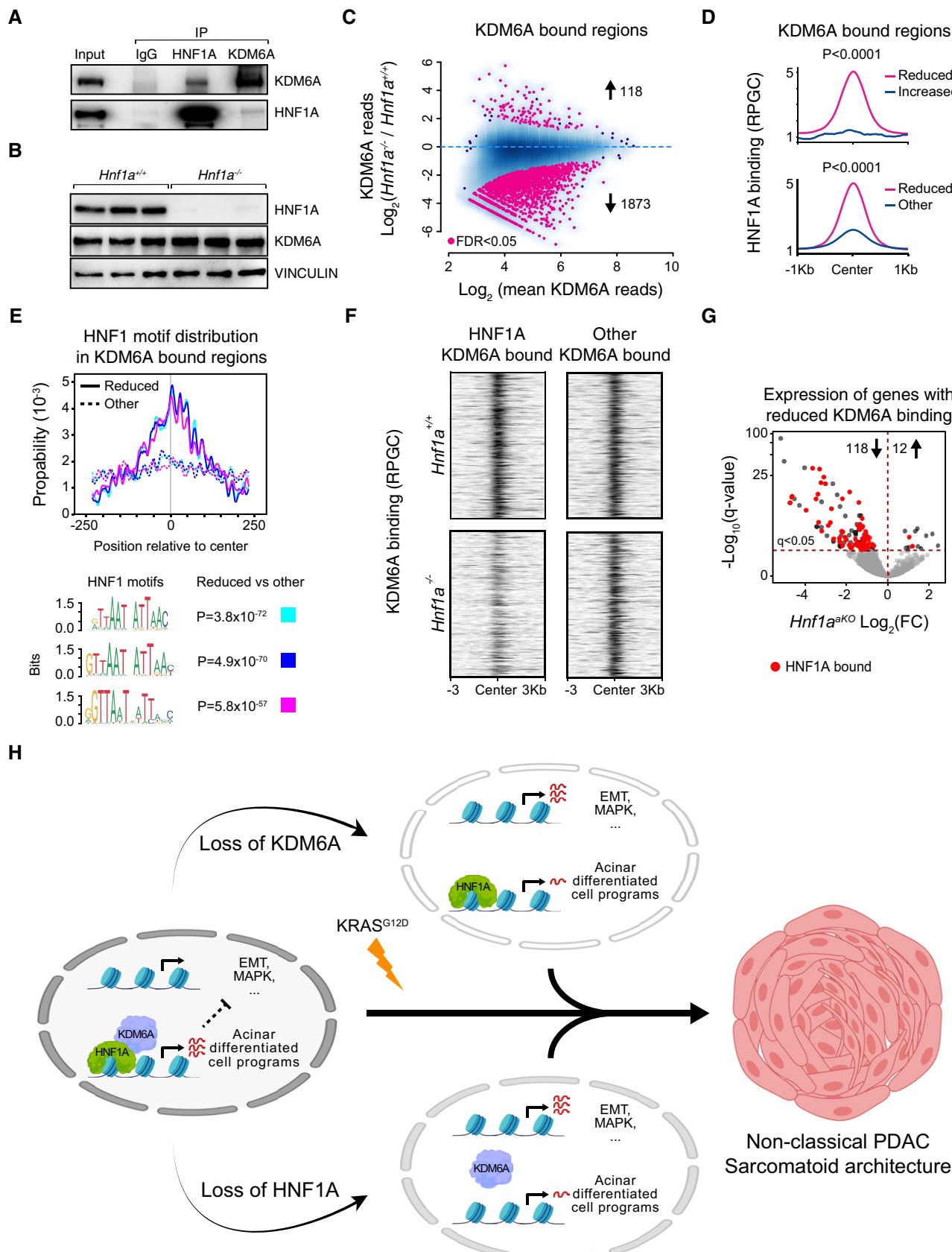

**Figure 7.**

◀

**Figure 7. HNF1A recruits KDM6A to activate transcription of its target genes.**

A  Co-immunoprecipitation of endogenous HNF1A and KDM6A followed by Western blot demonstrated that HNF1A is in the same complex as KDM6A.
B  Western blot showing loss of HNF1A and unchanged KDM6A in $Hnf1a^{-/-}$ pancreas.
C  Differential binding analysis of KDM6A in $Hnf1a^{-/-}$ versus wild-type pancreas. Pink dots below zero (1,873 sites) show regions with reduced KDM6A binding, and pink dots above zero (118 sites) are regions with increased binding at FDR < 0.05.
D, E  Regions that show reduced KDM6A binding in $Hnf1a^{-/-}$ chromatin are strongly bound by HNF1A and are highly enriched in HNF1 motifs. $P$-values in (D) were calculated with two-tailed Mann–Whitney U-test and in (E) with Fisher's exact test.
F  KDM6A binding is markedly reduced in HNF1A- and KDM6A-co-bound regions in $Hnf1a^{-/-}$ pancreas, but not in other KDM6A-bound regions.
G  Genes that loose KDM6A binding in $Hnf1a$-mutant pancreas are predominantly downregulated in $Hnf1a^{aKO}$ pancreas and are direct HNF1A target genes (red dots).
H  Summary model depicting that HNF1A recruits KDM6A to genomic binding sites, activating an acinar differentiation program that indirectly suppresses core oncogenic pathways. Defective HNF1A or KDM6A function results in failure of this shared program, with increased activity of pathways that, in the presence of $KRAS$ mutations, promote high-grade non-classical PDAC with sarcomatoid features.

(Yamagata *et al*, 1996), has a tumor-suppressive function in the exocrine pancreas. We further provide epigenomic, biochemical, and genetic evidence that mechanistically link *HNF1A* function in pancreatic exocrine cells to *KDM6A*, an established tumor suppressor (van Haaften *et al*, 2009). These results in turn indicate that the tumor-suppressive function of *KDM6A* is tightly linked to its role as a transcriptional co-regulator of epithelial differentiation. We demonstrate that HNF1A deficiency partially phenocopies KDM6A-driven tumorigenesis in mice, and show that *HNF1A/KDM6A* deficiency is a prominent feature of the genetic programs of non-classical human PDAC tumors. These findings, therefore, shed light on mechanistic underpinnings of emerging sub-classifications of PDAC subtypes.

Our analysis of *Kdm6a*-deficient *Kras*$^{G12D}$ mice is consistent with a recent report showing aggressive PDAC in *Kdm6a*-deficient *Kras* mutant mice (Andricovich *et al*, 2018), and extend those studies by defining the genetic programs that are directly regulated by KDM6A in the differentiated pancreatic exocrine cells that give rise to PDAC. The KDM6A-deficient transcriptome phenotype observed in non-tumoral pancreas was broadly consistent with that reported in tumors from *Kdm6a*-deficient *Kras* mutant mice (Andricovich *et al*, 2018). Our integrated analysis of genomic binding sites indicated that the positive regulatory effects of KDM6A (and HNF1A) on acinar differentiation genes are largely mediated through direct mechanisms. By contrast, genes associated with EMT, remodeling of the extracellular matrix, and oncogenic pathways that were upregulated in mutant cells were not frequently bound by HNF1A and KDM6A, whereas both proteins bound to known negative regulators of EMT. These findings strongly suggest that the suppression of oncogenic and EMT transcriptional programs is predominantly indirect.

Our studies, therefore, exposed a molecular mechanism whereby HNF1A recruits KDM6A to activate differentiation programs of pancreatic exocrine epithelial cells, thereby suppressing oncogenic and EMT pathways (Fig 7H). This mechanism aligns well with the observation that *Kdm6a*- and *Hnf1a*-deficient tumors in mice featured sarcomatoid architecture. We further showed that *HNF1A*- and *KDM6A*-dependent pancreatic programs are defective in non-classical PDAC tumors, variously defined as human basal, quasimesenchymal, or squamous molecular types (Collisson *et al*, 2011; Moffitt *et al*, 2015; Bailey *et al*, 2016; Cancer Genome Atlas Research Network, Electronic Address Aadhe, Cancer Genome Atlas Research N, 2017, Puleo *et al*, 2018). Interestingly, abnormalities in the *HNF1A/KDM6A*-dependent pancreatic program in human tumors were associated with either somatic *KDM6A* mutations, or to decreased expression of *KDM6A* or *HNF1A*, in consonance with the proposed molecular mechanism.

These findings indicate that the disruption of *HNF1A*- and *KDM6A*-dependent programs not only promotes the formation of PDAC, but is also instructive for the definition of PDAC subtypes. *HNF1A* is expressed in normal acinar cells, the presumed cell of origin of most PanIN precursors and PDAC (Kopp *et al*, 2012; von Figura *et al*, 2014), but not in duct cells. The results suggest that in the presence of other oncogenic events, defective *HNF1A*- or *KDM6A*-dependent gene regulation can promote the development of an undifferentiated high-grade PDAC cellular phenotype (Fig 7H). More generally, our results open avenues to study the precise manner in which *HNF1A*- and *KDM6A*-dependent regulation of epithelial differentiation programs fails and thereby contributes to the development of PDAC.

## Materials and Methods

### Animal studies

Animal experimentation was carried out in compliance with the EU Directive 86/609/EEC and Recommendation 2007/526/EC regarding the protection of animals used for experimental and other scientific purposes, enacted under Spanish law 1201/2005.

The *Kdm6a*$^{LoxP}$ (Shpargel *et al*, 2012), *Pdx1*$^{Cre}$ (Hingorani *et al*, 2003), *LSL-Kras*$^{G12D}$ (Jackson *et al*, 2001), *Hnf1a*$^{-/-}$ (Lee *et al*, 1998), and *Ptf1a*$^{Cre}$ (Kawaguchi *et al*, 2002) mouse lines have been described.

To generate a conditional *Hnf1a* allele, a floxed *Hnf1a* exon 2 was generated in C57Bl/6N JM8.F6 embryonic stem cells (ES cells) (Pettitt *et al*, 2009) by homologous recombination. Briefly, ES cells were electroporated with a targeting plasmid construct linearized by PmeI and MluI, containing the floxed exon and a PGK/Neomycin selection cassette flanked by FRT recombination sites, and then selected with Geneticin. Several correctly targeted ES cell clones were identified by Southern blot using external probes detecting a 9.2 kb 5′ ScaI fragment, an 8.7 kb 3′ BamHI fragment from the recombined allele, and a neo probe detecting a unique 11.7 kb SpeI Neomycin fragment indicating no additional construct integrations. The presence of the distal LoxP site, upstream of exon 2, was confirmed by PCR using the primers indicated in Table EV1. Correctly targeted ES cell clones were injected into C57BL/6BrdCrHsd-Tyrc morulae (E2.5) to create chimeric mice that transmitted the recombined allele through the germ line. The PGK/Neo cassette was excised by crossing

heterozygous mice with Tg.CAG-Flp mice (Rodriguez *et al*, 2000), which express the FLPe recombinase in the germ line to create the *Hnf1a* conditional knock-out allele. Lines with floxed alleles without Cre, Cre lines without floxed alleles, *Pdx1^{Cre};LSL-KRAS^{G12D}* or wild-type littermates served as controls. Gender was considered as a variable and included in the study design. Genotype was confirmed by PCR using the primers provided in Table EV1.

Prior to decapitation, mice were anesthetized using isoflurane (Zoetis), and pancreas was collected quickly and placed in paraformaldehyde 4% (Sigma-Aldrich) for paraffin embedding or processed for RNA isolation.

## Statistical methods

Statistical tests are described throughout the materials and method section and mentioned in figure legends and main text where appropriate. Two-tailed Student's *t*-test was used to compare two sets of data from qPCR experiments. For comparisons between two independent groups of genome-wide data with continuous variables, we used two-sided Mann–Whitney *U*-test. Non-parametric Kruskal–Wallis test was used to assess more than two groups with dependent values, including RNA-seq and microarray data. Fisher's exact test was used to determine whether there were nonrandom associations between two categorical variables. Data are presented as means with their standard deviations, or as box plots with median, IQR, and whiskers that extend to data points that are < 1.5× IQR away from the first and third quartile. Outliers are defined as values larger than the third quartile by at least 1.5× IQR, or smaller than the first quartile by at least 1.5× IQR.

## Histology, TMA preparation, and Immunohistochemistry

Fixation of mouse pancreas was performed using paraformaldehyde 4% (Sigma-Aldrich) or 4% buffered formalin overnight at 4°C followed by dehydration using increasing concentrations of ethanol. Dehydrated samples were embedded in paraffin blocks and cut in 4-µm sections. Sections were then deparaffinized and rehydrated. For hematoxylin and eosin analysis, the pancreata were stained, dehydrated, mounted with DPX (Panreac), and photodocumented using optical microscopy (Olympus BX41TF).

Tissue samples for tissue micro arrays (TMAs) were obtained from patients who underwent pancreatic resection for PDAC (*n* = 223) at the Department of Surgery of the University Hospital of Duesseldorf, Germany. The samples were fixed in 4% formalin and subsequently routinely embedded in paraffin. Three tumor samples per case (two samples from the tumor center, one from the periphery) with a 1-mm core size were selected and assembled into the TMA (Manual Tissue Arrayer MTA-1, Beecher Instruments, Inc., Sun Prairie, WI, USA). The use of human tissue samples was approved by the local ethics committee at the University Hospital of Duesseldorf, Germany (study number 5387). The TMAs were then cut in 2-µm sections for subsequent procedures.

Immunohistochemistry on mouse pancreas or TMAs was performed according to manufacturer's instructions using VectaStain® ABC HRP Rabbit IgG kit (Vector Laboratories, Peterborough, UK). Epitope retrieval was achieved by pretreatment with boiling citrate buffer (10 mM citric acid, pH6) for 15 min using a pressure cooker. Endogenous peroxidase and protein blocking was performed with 3% $H_2O_2$ diluted in PBS for 10 min and with 1% BSA, 10% normal goat serum (Abcam, Cambridge, UK), and 0.1% Triton X-100 (Merck KGaA, Darmstadt, Germany) for 60 min. Anti-HNF1A and anti-KDM6A stainings were performed at a dilution of 1:250 (Anti-HNF1A, Abcam ab204306, Cambridge, UK), 1:200 (Anti-HNF1A, Cell Signaling Technology, 89670, Leiden, The Netherlands), and 1:100 (Anti-UTX, Cell Signaling Technology 33510S, Denver, USA), respectively. Diaminobenzidine (Abcam ab64238, Cambridge, UK) was used as chromogen to visualize protein expression, counterstaining was achieved with Mayer's hemalum solution (Merck KGaA, Darmstadt, Germany).

## Analysis of TMAs

The semi-quantitative analysis of the stained sections was done by light microscopy according to the immunoreactive score (IRS) of Remmele and Stegner (Remmele & Stegner, 1987). Both the staining intensity and the number (%) of positive stained cells were evaluated for each case. If more than one tumor core was available in one case, a mean value was calculated. Staining intensity was scored as no staining (0), weak staining (1), moderate staining (2), or strong staining (3). The proportion of stained cells was scored as follows: no staining = 0, 1–9% = 1, 10–50% = 2, 50–80% = 3, > 80% = 4. The IRS was calculated by multiplication of the intensity value and the percentage value, resulting in a score between 0 and 12. Finally, IRS results were classified into four groups as follows: 0–1 = negative (group 0), 2–3 = weak (group 1), 4–8 = moderate (group 2), 9–12 = strong (group 3). Each tumor case was graded according to UICC and WHO classifications. Two cases with undetermined grade (Gx) were omitted from the analysis. The rest of the cases were either scored as G2 or G3 grades. Contingency table analysis and Chi-square test were used to evaluate the association between IRS classifications and tumor grades.

## Immunofluorescence

Immunofluorescence analysis of murine pancreata was performed on paraformaldehyde-fixed paraffin-embedded tissue cut into 4-µm sections, as described with modifications (Solar *et al*, 2009). Antigen unmasking was performed boiling sections on 10 mM citrate buffer pH 6 for 10 min, and tissue was permeabilized with PBS 0.5% Triton X-100 (Sigma). Blocking was performed by 1-h incubation in humid chamber with 3% of normal donkey serum (Vendor) in blocking solution (DAKO). Slides were incubated in humid chamber with primary antibodies overnight at 4°C and washed in PBS 0.2% Triton X-100. Slides were incubated with secondary antibodies for 1 h in humid chamber at room temperature, washed for 5 min in PBS 0.2% Triton X-100, and mounted using fluorescence mounting medium (DAKO). Stained slides were analyzed using a confocal laser scanning microscope (Leica CTR6500). Primary antibodies used are indicated in Table EV2.

## Western blotting and co-immunoprecipitation

All protein isolation steps were performed at 4°C or on ice, and protein concentrations were assessed with the Bradford assay

(Bio-Rad). Protein from total pancreas lysates was isolated from a small fragment of mouse pancreas and minced in lysis buffer (20 mM Tris–HCl, pH 7.5, 150 mM NaCl, 1 mM $Na_2EDTA$, 1 mM EGTA, 1% Triton X-100, 2.5 mM sodium pyrophosphate, 1 mM b-glycerophosphate, 1 mM $Na_3VO_4$, 1 µg/ml leupeptin containing 1× phosphatase inhibitor cocktail, Sigma-Aldrich, and 4× EDTA containing complete protease inhibitor cocktail, Roche). Lysates were freeze-thawed twice, cleared at 15,000 $g$ for 15 min, and then, the supernatant was recovered. For Western blots, lysate supernatants were denatured in 2× NuPAGE LDS sample buffer (Thermo Fisher) containing 0.1 M DTT at 95°C for 5 min. Equal amount of lysates were loaded in NuPAGE 4–12% Bis-Tris Protein Gels (Life Technologies) and run at 150V for 90 min. Gels were transferred to 0.2-µm pore PVDF membrane (Life Technologies) and blocked with 5% fat-free milk powder for 60 min and probed overnight (ON) with primary antibody. Horseradish peroxidase conjugated secondary antibodies were used at 1:10,000 in BSA 5% and incubated 1 h at RT. The blot was developed using ECL detection reagent (Amersham ECL, GE Healthcare). The pancreatic acinar cell line 266-6 (ATCC CRL-2151) was grown in DMEM (Thermo Fisher) with 10% fetal calf serum, 1% pen/strep, and passaged twice per week. For co-immunoprecipitation, the cells were washed in ice-cold PBS and resuspended in hypotonic lysis buffer (10 mM HEPES, pH 7.9, with 1.5 mM $MgCl_2$ and 10 mM KCl). After plasma membrane disruption, the lysate was centrifuged at 4,000 $g$ for 5 min and the nuclei lysed for 20 min with gentle shaking in nuclear extraction buffer (20 mM HEPES, pH 7.9, with 1.5 mM $MgCl_2$, 0.42 M NaCl, 0.2 mM EDTA, and 25% (v/v) Glycerol) before removing cell debris by centrifugation at 15,000 $g$ for 15 min. The cleared nuclear lysate was equilibrated to 150 mM NaCl and pre-cleared with Dynabeads Protein G (Thermo Fisher) for 1 h at 4°C with rotation, and 4 mg of extract was incubated overnight with 2 µg of anti-KDM6A, anti-HNF1A, or normal rabbit IgG. Immune complexes were precipitated with 20 µl Dynabeads for 2 h and washed four times with IP wash buffer (10 mM Tris–HCl, pH 8, 150 mM NaCl, 0.1% NP-40, and 1 mM EDTA) supplemented with protease inhibitor. Proteins were eluted with 2× NuPAGE LDS sample buffer (Thermo Fisher) containing 0.1 M DTT at 95°C for 5 min and analyzed by Western blotting.

## RNA isolation

For gene expression profiling, we implemented a method for purification of intact pancreatic RNA according to a modified guanidinium salts method (MacDonald *et al*, 1987). Briefly, a small piece of pancreatic tissue was immediately homogenized with a Polytron (VWR) in pre-cooled 4 M guanidinium thiocyanate with 112 mM beta-mercaptoethanol, centrifuged at 5,000 $g$ for 5 min at 4°C to remove insoluble debris. RNA was precipitated from the supernatant with pre-cooled 75% ethanol, 0.1M potassium acetate, pH 5.5, and 75 mM acetic acid at −20°C for 2 h. The precipitate was pelleted by centrifugation at 10,000 $g$ for 10 min at 4°C and resuspended at room temperature in 7.5 M guanidinium HCl and 10.5 mM beta-mercaptoethanol. The RNA was re-precipitated twice with 0.1 M potassium acetate, pH 5.5, and 50% ethanol to remove residual RNases, followed by purification with TRI Reagent RNA Isolation Reagent (Sigma-Aldrich). The concentration and quality of RNA was measured with a NanoDrop spectrophotometer (ND-1000,

Thermo Scientific) and an Agilent 2100 Bioanalyzer. RNA integrity numbers ranged from 7.8 to 9.3.

Alternatively, we used a previously described protocol (Cobo *et al*, 2018).

## RNA-seq

One µg of total RNA was used to make RNA-seq libraries using the Truseq Stranded mRNA Sample Prep Kit (Illumina) following manufacturer's instructions and sequenced by an Illumina HiSeq2500 platform with single-end reads of 50 bases. Conversion to FASTQ read format was done using Illumina's bcl2fastq algorithm. Four or three pancreases from, respectively, 4-day-old or adult female mice of each genotype were analyzed: $Kdm6a^{LoxP/LoxP}$ and $Pdx1^{Cre}$; $Kdm6a^{LoxP/LoxP}$ or $Ptf1a^{Cre}$ and $Ptf1a^{Cre};Hnf1a^{LoxP/LoxP}$. Raw RNA-seq reads were aligned to the mouse transcriptome (a combined build of cDNA and ncRNA from Mus musculus v.GRCm38.p5, release 87) and quantified using Salmon v0.7.2 (Patro *et al*, 2017), with the following parameters: salmon quant –gcBias, – libType A, and –fldMean and –fldSD. The latter two were set to the average size and standard deviation of the fragment length distribution of the given library, respectively. The counts were used in DEseq2 (v1.14.1) (Love *et al*, 2014) with R (v3.3.3) (RStudio, 2015) for normalization and identification of significant differential expression ($q < 0.05$) between controls ($n = 4$) and mutants ($n = 4$). Only genes with log2 (basemean) > 2.5 were retained for downstream analysis. For visualization, STAR (v2.3.0) (Dobin *et al*, 2013) was used to align reads to the mouse genome assembly GRCm38.p5 (mm10) and resulting BAM files were converted to bigwig with reads per genomic content (RPGC) normalization by Deeptools (v2.4.2) (Ramirez *et al*, 2016).

## ChIP

ChIP experiments were performed on pancreas from 4-day-old or 3-week-old female mice. A Polytron was used to quickly mince 1–2 dissected pancreas in PBS with protease inhibitors followed by crosslinking with 1% formaldehyde for 10 min at room temperature on a rotator and then quenched with 0.125 M glycine for 5 min. Tissue pieces were washed in PBS and lysed in 130 µl lysis buffer (2% Triton X-100, 1% SDS, 100 mM NaCl, 10 mM Tris–HCl, pH 8.0, 1 mM EDTA) with protease inhibitor on ice for 10 min and finally resuspended with a 30G needle syringe. The chromatin preparation was then sonicated to fragments enriched in the size range of 150–500 bp using S220 Focused Ultrasonicator (Covaris) and centrifuged at 18,000 $g$ for 10 min at 4°C to pellet insoluble material. The supernatant was incubated for 1 h with 350 µl RIPA-LS buffer (10 mM Tris–HCl, pH 8.0, 140 mM NaCl, 1 mM EDTA, pH 8.0, 0.1% SDS, 0.1% Na-Deoxycholate, 1% Triton X-100) with protease inhibitor and 20 µl Dynabeads Protein G for pre-clearing. After preserving 1% as an input sample, the pre-cleared chromatin was incubated with 2 µg antibody, 50 µl 10% BSA, and 5 µl tRNA (10 mg/ml) overnight at 4°C with rotation. Immune complexes were retrieved with 20 µl BSA blocked Dynabeads Protein G for 2 h and washed 2× with RIPA-LS, 2× with RIPA-HS (10 mM Tris–HCl, pH 8.0, 500 mM NaCl, 1 mM EDTA, pH 8.0, 0.1% SDS, 0.1% Na-Deoxycholate, 1% Triton X-100), 2× with RIPA-LiCl (10 mM Tris–HCl, pH 8.0, 250 mM LiCl, 1 mM EDTA, pH 8.0, 0.5% IGEPAL, 0.5% Na-Deoxycholate), and 1× with 10 mM Tris, pH 8.0.

ChIP-seq libraries were prepared directly on the bead-bound chromatin according to a ChIPmentation procedure that is optimized for low cell number samples (Schmidl *et al*, 2015). Briefly, beads were resuspended in 25 µl of tagmentation reaction buffer (10 mM Tris, pH 8.0, 5 mM MgCl$_2$, 10% v/v dimethylformamide) containing 1 µl Tagment DNA Enzyme from the Nextera DNA Sample Prep Kit (Illumina) and incubated at 37°C for 10 min followed by 2× washing in RIPA-LS and TE (10 mM Tris–HCL, pH 8.0, 1 mM EDTA, pH 8.0). The DNA was then de-crosslinked and eluted from the beads in ChIP elution buffer (10 mM Tris–HCL, pH 8.0, 5 mM EDTA, pH 8.0, 300 mM NaCl, 0.4% SDS) with proteinase K at 55°C for 1 h and 65°C overnight. ChIP DNA was purified with Qiagen MinElute kit (QIAGEN), and enrichment cycles for library amplification were assessed by qPCR. The libraries were PCR amplified with KAPA HiFi Hotstart Ready mix (Sigma-Aldrich) and barcoded Nextera custom primers (Schmidl *et al*, 2015) and finally size-selected (250–350 bp) using Agencourt AMPure XP beads (Beckman Coulter) and validated using the Agilent High Sensitivity DNA Kit with Agilent 2100 Bioanalyzer. Equimolar quantities of libraries were combined for multiplexing to obtain 40 million reads per library. ChIP-seq libraries were sequenced using a NextSeq platform with single-end reads of 75 bases.

For ChIP-qPCR, DNA was de-crosslinked and eluted from the beads in ChIP elution buffer (10 mM Tris–HCL, pH 8.0, 5 mM EDTA, pH 8.0, 300 mM NaCl, 0.4% SDS) with proteinase K at 55°C for 1 h and 65°C overnight, and purified by phenol chloroform extraction according to standard procedures. Quantitative PCR was then performed on a QuantStudio 6 Flex Realtime PCR machine (Applied Biosystems) using GoTaq qPCR reagent (Promega). Fold enrichments were visualized relative to input and negative control regions using primers available in Table EV1.

### ChIP-seq alignment and peak calling

ChIP-seq reads were aligned to the mouse genome (*M. musculus*, UCSC mm10) using Bowtie2 followed by exclusion of mapping quality scores < 30 by Samtools (v1.2) (Li *et al*, 2009) and removal of duplicate reads with Picard (v2.6.0) and blacklisted regions with Bedtools (v2.13.3) (Quinlan & Hall, 2010). The processed BAM files were converted to bigwig files using Deeptools. Peaks for histone marks were called with MACS2 (v.2.1.0) with settings: –extsize = 300 –q 0.05 –keep-dup all –nomodel –broad and for KDM6A and transcription factors with MACS1.4 (v1.4.2) (Zhang *et al*, 2008) using default parameters and $P < 10^{-10}$. Enriched regions were scored against matching input libraries. All experiments were done as biological replicates. For histone marks and KDM6A ChIPs, only enriched regions that overlap in replicates were retained as consistent peaksets. Publically available datasets were processed identically. For HNF1A ChIPs, peaks from two replicates that overlap by at least one base were merged and replaced with a single peak. Library information of public and internal datasets is provided in Table EV3.

### Integrative analysis

Active promoters were defined as consistent H3K27ac peaks occurring within 1 kb of an annotated transcription start site (GENCODE GRCm38.p5), and all remaining peaks were considered as active enhancers. HNF1A and KDM6A peaks were annotated with HOMER

(v4.10.3) (Heinz *et al*, 2010) to define TSS-proximal, TSS-distal, intronic, exonic, and intergenic regions, and were assigned to genes using the "closest" function in Bedtools with default parameters. Fisher's exact test in R was used to define enrichment of HNF1A- and KDM6A-bound regions and active promoter and enhancer regions, or differentially expressed genes in the *Hnf1a*$^{aKO}$ and *Kdm6a*$^{pKO}$ datasets. Expressed genes (Log2 TPM > 2.5) were used as background. Aggregation plots were calculated with Deeptools. Briefly, all KDM6A-bound regions were extended ± 10 kb or ± 5 kb from the center of the peaks for analysis of H3K27me3 or H3K27ac signals, respectively. The resulting regions were then divided into 10-bp bins, and the mean value in each bin was calculated based on the normalized values given in the bigwig files for the histone marks. To assess the relationship between functional KDM6A binding sites and KDM6A-dependent histone modification changes, the Log$_2$ fold change of the averaged normalized H3K27me3 signals was plotted against the Log$_2$ fold change of H3K27ac signals in each KDM6A-bound region. Differential binding of KDM6A and HNF1A was analyzed with DiffBind (v2.6.6) (Ross-Innes *et al*, 2012) using the DEseq2 method without control input read counts. Differential peaks with FDR < 0.05 were defined as significant.

### Functional annotation and gene set enrichment analysis (GSEA)

The ENRICHR tool (Chen *et al*, 2013; Kuleshov *et al*, 2016) was used to functionally annotate differentially expressed genes. Significantly enriched signatures were identified using Bonferroni corrected $P < 0.01$. The GSEA pre-ranked analysis (Subramanian *et al*, 2005) was used to determine whether predefined sets of genes show significant concordant differences with gene expression changes in RNA-seq datasets. For GSEA, we used gene sets from the Molecular Signature Database (Liberzon *et al*, 2015), custom-made sets of differentially expressed genes from our dataset, gene sets of pancreatic cell types (Muraro *et al*, 2016), and human PDAC (Collisson *et al*, 2011; Moffitt *et al*, 2015; Bailey *et al*, 2016). All were analyzed with the parameters: numbers of permutations = 10,000 and scoring scheme = weighted.

### Motif analysis

To find enriched *de novo* transcription factor DNA-binding profiles, we used HOMER (Heinz *et al*, 2010) and searched for 6-, 8-, 10-, and 12-bp sized motifs in ± 250-bp regions surrounding the center of KDM6A-bound regions within promoters and enhancers that were functionally associated with transcriptional changes. As background sequences, we used all H3K27ac regions. DNA motifs were annotated with HOMER and retrieved if the HOMER score was more than 0.7. Centrimo from the MEME suite (Bailey & Machanick, 2012) was used to compute enrichment of known HNF1A motifs in HNF1A-bound regions (Fig EV1I) and in KDM6A-bound regions (Fig 7E). Position weight matrices for HNF1 were retrieved from JASPAR (Sandelin *et al*, 2004) (http://jaspar.genereg.net/) and HOCOMOCO (Kulakovskiy *et al*, 2018) (http://hocomoco11.autosome.ru/) databases.

### Data visualization

Bigwig files from RNA-seq and DNA-seq experiments were visualized in the Genome Browser (http://genome.ucsc.edu/). R packages

were used to create boxplots (BoxPlotR) (Spitzer *et al*, 2014), violin-plots (PlotsOfData) (Postma & Goedhart, 2019), and bar and scatter plots (ggplot2) (Wickham, 2009). Enrichment plots from GSEA were drawn by Genepattern (Reich *et al*, 2006). Aggregation plots and heatmaps of ChIP experiments were drawn with Deeptools. GraphPad Prism 6 was used for bar graphs, calculation of contingency tables and piecharts. To draw heatmaps, we used Morpheus (https://software.broadinstitute.org/morpheus).

### GTEx analysis

Quantile-normalized read counts of 328 pancreas samples from GTEx version 8 dataset (GTEx_Analysis_2017-06-05_v8_RNASeQCv1.1.9_gene_tpm.gct.gz) were retrieved from https://gtexportal.org/home/datasets. Samples were ranked according to *HNF1A* levels, and fold differences between bottom and top expression deciles ($n = 33$ per group) were calculated. We then examined differential expression of orthologous genes in *Hnf1a*[aKO] versus control pancreas. A random list of 717 genes controlled for similar expression levels was used for comparison.

### Analysis of human PDAC genomic data

#### Data acquisition and preprocessing

RSEM normalized RNA-seq data from the TCGA-PAAD cohort was downloaded from the Firehose browser (https://gdac.broadinstitute.org/). Molecular subtypes and purity class were assigned to each sample as defined by the Cancer Genome Atlas Research Network (Cancer Genome Atlas Research Network, Electronic Address Aadhe, Cancer Genome Atlas Research N, 2017). Samples with high purity (76 samples) were retained for further analysis. Quantile-normalized background-adjusted array expression data from the ICGC-PACA-AU cohort were downloaded from the ICGC data portal (https://dcc.icgc.org/). Probe sets (ILLUMINA HumanHT 12 V4) were assigned to gene names. Multiple probe sets mapping to the same gene were collapsed using the collapseRows algorithm from the WGCNA package (Langfelder & Horvath, 2008) keeping the gene-probe set combination with highest variance across all samples. Purity class was assigned to each sample according to Bailey *et al* (2016) retaining 121 high-purity samples for further analysis.

#### Consensus clustering

Samples from the ICGC-PACA-AU study were classified into molecular subtypes based on signature genes from Collisson *et al* (2011), Moffitt *et al* (2015), or Bailey *et al* (2016). Out of the 62 PDA assigner genes identified by Collisson *et al*, 57 matched with our preprocessed data. We also selected 50 genes with the highest gene weights for the basal and classical PDAC subtypes (100 genes in total) from Moffitt *et al*'s gene factorization analysis, out of which 94 genes matched with our data. From Bailey *et al*'s 613 differentially expressed genes across their defined classes (Squamous, ADEX, Immunogenic, and Pancreatic Progenitor), we retained 457 matching genes. We then applied consensus clustering to the ICGC-PACA-AU data with the signature genes from the three studies using ConsensusClusterPlus (CCP) v.1.24.0 (Wilkerson & Hayes, 2010). Each gene expression profile was first $Log_{10}$-transformed and median centered. We next performed 1,000 iterations

of CCP using Pearson correlation as the distance metric, partitioning around medoids, and a random gene and sample fraction of 90% in each iteration. We verified that the obtained groups and their gene expressions reflected the up and down relationships for each gene in each group described in the three studies. We next used human orthologs of differentially expressed genes from our study in *Hnf1a*-deficient mouse pancreas to cluster the TCGA-PAAD and ICGC-PACA-AU cohorts. We applied consensus clustering with non-negative matrix factorization, using Pearson distance metrics and 2,000 descent iterations (Kuehn *et al*, 2008), and thereby identified a group of samples with similar expression signature to the *Hnf1a*[aKO] pancreas. To generate expression heatmaps, we identified genes that were differentially expressed across the clusters (FDR < 0.05) with a SAM multiclass analysis (samr v.2.0) (Tusher *et al*, 2001), $Z$-score transformed each row of the matrix, then used Morpheus (https://software.broadinstitute.org/morpheus) to cluster only the rows, using a one minus Pearson correlation distance metric, average linkage method, and Hierarchical clustering.

To identify tumors with most pronounced HNF1A-deficient function, we first defined a gene set containing 106 human orthologs of HNF1A-bound genes that were downregulated in *Hnf1a*[aKO] mouse pancreas. We then interrogated that behavior of this gene set in every tumor sample. This was carried out by GSEA (see also GSEA subheading), using the HNF1A-dependent gene set, and testing the enrichment in each tumor's gene lists rank-ordered by differential expression of all genes in the tumor versus the median expression of all genes across all samples. This analysis revealed 3 groups of tumor samples: (i) HNF1A loss of function (LoF) with NES < 0, $P < 0.05$, (ii) *Control 1* with NES < 0, $P > 0.05$, and (iii) *Control 2* with NES > 0.

#### KDM6A mutations

Information on *KDM6A* mutations was retrieved from the ICGC data portal (https://dcc.icgc.org) and Bailey *et al* (2016). 15 samples in the high-purity ICGC-PACA-AU cohort had *KDM6A* mutations, which we separated into three groups according to mutation types defined by ICGC: (i) Deletions (small ≤ 200 bp), (ii) Substitutions (single base), (iii) Insertions (small ≤ 200 bp. The fourth group, (iv) Structural Variants, was from Bailey *et al* Mutations defined by ICGC as having "high" functional impact in (1–3) and having a "loss-of-function" consequence in (iv) were all frame-shift mutations. We considered those mutations as being functional deleterious and defined samples with such genetic alterations in *KDM6A* as *KDM6A* loss-of-function (LoF) tumors. The rest were defined as samples with *KDM6A* mutations of unknown consequence. Significantly mutated genes (SMGs) were from Bailey *et al* (2016) SMG analysis using Intogen, Mutsig, and HOTNET. Genes considered significantly mutated were significant in > 1 analysis.

## Data availability

RNA-seq and ChIP-seq data sets generated here are available in the ArrayExpress database at EMBL-EBI (www.ebi.ac.uk/arrayexpress) under accession numbers E-MTAB-7944 and E-MTAB-7945.

**Expanded View** for this article is available online.

## Acknowledgements

We thank I. Cobo, S. Paliwal, and members of the Ferrer laboratory for valuable discussions and Sonia Corral Leal for drawing and advising the design of the video synopsis. We thank the Parc de Recerca Biomèdica de Barcelona and University of Barcelona School of Medicine animal facilities, Center of Genomic Regulation and Imperial College London Genomics Units, and the Imperial College High Performance Computing Service. This research was supported by the National Institute for Health Research (NIHR) Imperial Biomedical Research Centre. Work was funded by grants from the Wellcome Trust (WT101033 to J.F.), Medical Research Council (MR/L02036X/1 to J.F.), European Research Council Advanced Grant (789055 to J.F.), Ministerio de Ciencia e Innovación (BFU2014-54284-R, RTI2018-095666-B-I00 to J.F., SAF2011-29530 and SAF2015-70553-R to F.X.R.) and RTICC from Instituto de Salud Carlos III (RD12/0036/0034, RD12/0036/0050) to F.X.R. M.K. was supported by a Juvenile Diabetes Research Foundation postdoctoral fellowship (3-PDF-2014-192-A-N). I.M. was supported by a Fellowship from Fundació Bancaria La Caixa (ID 100010434) (grant number LCF/BQ/ES18/11670009). Work in CRG was supported by the CERCA Programme, Generalitat de Catalunya, and support from Ministerio de Ciencia e Innovación to the EMBL partnership. Work at CRG and CNIO was supported by Centro de Excelencia Severo Ochoa grants SEV-2012-0208, SEV-2016-0510.

## Author contributions

MK and JF conceived the study. MK, MAM, and EB analyzed mouse phenotypes. AB designed the Hnf1a mutant allele that was generated by SO. MK and IM performed bioinformatic analysis. NP and VG maintained mouse colonies. KBS and TM created the Kdm6a mutant model. EV provided the LSL-KrasG12D mice and MV provided samples. IE carried out mouse histopathological analysis. IE, LH, MS, SAS, and WTK analyzed TMAs. FXR provided expertise and resources for conducting experiments during the revision phase. MK, EB, FXR, IE, and JF wrote the manuscript with input from all authors.

## Conflict of interest

The authors declare that they have no conflict of interest.

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
