## [Review Process File · The EMBO Journal]

HNF1A recruits KDM6A to activate differentiated acinar cell programs that suppress pancreatic cancer

Mark Kalisz, Edgar Bernardo, Anthony Beucher, Miguel Angel Maestro, Natalia del Pozo, Irene Millán, Lena Haeberle, Martin Schlenz, Sami Alexander Safi, Wolfram Trudo Knoefel, Vanessa Grau, Matías de Vas, Karl B. Shpargel, Eva Vaquero, Terry Magnuson, Sagrario Ortega, Irene Esposito, Francisco X. Real, Jorge Ferrer

Review timeline:

Submission date:	1st Jul 2019
Editorial Decision:	22nd Jul 2019
Revision received:	5th Dec 2019
Editorial Decision:	8th Jan 2020
Revision received:	2nd Feb 2020
Accepted:	7th Feb 2020

Editor: Daniel Klimmeck

Transaction Report:

1st Editorial Decision

22nd Jul 2019

Thank you for the submission of your manuscript (EMBOJ-2019-102808) to The EMBO Journal. Your manuscript has been sent to three reviewers, and we have received reports from all of them, which I enclose below.

As you will see, the referees acknowledge the potential interest and novelty of your results, although they also express a number of issues that will have to be addressed before they can support publication of your manuscript in The EMBO Journal. In more detail, referee #2 states that a number of concerns about additional experiments and controls are required to consolidate your findings and further investigate interdependence of HNF1A and UTX in mouse and human. Referee #1 asks you to explore the pathways and signaling components downstream of HNF and/or UTX in greater depth. In addition, the reviewers raise a number of issues related to methods annotation, data representation, statistics and appropriate citation of literature references that would need to be conclusively addressed to achieve the level of robustness and clarity needed for The EMBO Journal.

I judge the comments of the referees to be generally reasonable and given their overall interest, we are happy to invite you to revise your manuscript experimentally to address the referees' comments.

REFeree REPORTS:

Referee #1:

This study focuses on the tumor-suppressive function of the transcription factor HNF1A in the exocrine pancreatic tissue. The authors studied HNF1A-deficiency alone and in combination with KrasG12D PDAC model. Moreover, the pancreas-specific Hnf1a null mutations were compared to the histone demethylase UTX/KDM6A deficient mutations in PDAC mouse models. Both mutations

synergize with KrasG12D leading to PDAC with partially overlapping features. Based on transcriptomics, epigenomic and biochemical studies the authors suggest a molecular mechanism whereby HNF1A is required for the recruitment of UTX to its genomic targets in acinar cells. This recruitment remodels the enhancer landscape of acinar cells and activates a differentiation program and, indirectly, suppresses oncogenic and epithelial-mesenchymal transition genes. Finally, these observations were extended to human PDAC transcriptome data and defined a subset of non-classical PDAC human tumors that exhibit HNF1A/UTX-deficient transcriptional programs.

Overall, this is a very interesting study that sheds light on how epigenetic modifiers, such as UTX, work together with lineage-specific transcription factors, such as HNF1A, in the context of pancreatic differentiation and cancer.

All the genetics, transcriptome and biochemical data are very convincing and support well the proposed model of a direct positive regulatory control of UTX through HNF1A on pancreatic differentiation. By contrast, the evidence for an indirect negative control on EMT and oncogenic pathways is tenuous - this could be expanded or better discussed by the authors.

Additional comments:

- The authors should further explain or discuss why a non-classical PDAC sarcomatoid architecture is associated with HNF1A/UTX-deficient molecular signature. Is it because of the ECM remodelling or the EMT genes dysregulation?

- Looking at the histology in Fig.1A and Fig. S5i, I wonder if there is any consequence in the pancreatic tissue-architecture and lobule organization in the HNF1A- and UTX-single KO animals. If this can be excluded, then the micrographs shown in Fig.1a and Fig. S5i are not representative and should be replaced.

- Is it known where UTX is expressed in normal pancreatic tissue?

Referee #2:

This is an elegantly done manuscript by Kalisz and colleagues supporting a role for the functional interaction between UTX and HNF1A in the development of Kras-induced pancreatic tumor initiation. The investigational team used a set of state-of-the-art animal models and genome wide studies to evaluate the transcriptional programs used by this complex in the regulation of gene expression during pancreatic transformation. The data and writing are clearly presented, however, additional results are needed to fully determine the impact of the UTX-HNF1A complex in the regulation of pancreatic carcinogenesis. Specifically, the nature of the interaction, the role in human models as well as lack of key controls.

Specific comments:

- 1) The effect HNF1A loss on the expression UTX should be evaluated as the magnitude of the drop in UTX enrichment is difficult to account based on the number of HNF1A targets. The differences could be explained by a decreased expression of UTX and not lack of recruitment to target genes.
- 2) The validation of key findings (cell growth, target gene regulation and interaction - IP and CHIP) should be done in pancreatic human cells (KRAS-transformed and non-transformed). This will enhance the disease significance of the study.
- 3) ChIP-seq using genome browser track is not ideal, at least some key genes and chromatin landscaping mapping should be validated by ChIP-PCR in tissue or cell lines derived from the WT and KO models. Also, the levels of H3K4me should be evaluated in both WT and KO cells.
- 4) The levels of Kras activation should be evaluated in both WT and KO models by examining the levels of pERK.
- 5) H/E of the Pdx1-HNF1A KO should be shown (high and low magnifications) and low magnification of the UTX KO, the IF images are insufficient to evaluate the impact of the KOs on the histomorphology of the pancreas. The positive signal (Sup Fig 1C) should be explained, it seems that HNF1A KO still has some acinar signal?
- 6) It will be ideal (but not critical) that the TMAs are co-stained for UTX and the UTX mutational status should be included.

- 7) Validation of HNF1A and UTX targets by ChIP-PCR and qPCR is a must. This should be done on tissue or cell lines derived from the mouse models. It is important to define the co-occupancy of both molecules in a selected group of target genes. Also, it will be important to determine if the recruitment to target genes is induced by mutant KRAS?
- 8) The requirement for UTX for the HNF1A-dependent transcription should be examined using RNAi and overexpression tools in cell lines models. So far, the authors have shown associations to support this event, not a cause-effect.
- 9) The histological and transcriptional differences with Cancer Cell article from Tzatzos's group should be discussed.
- 10) The authors should evaluate (in an in vitro system) the impact of some of the genes commonly regulated the transcriptional program regulated by HNF1A-UTX in KRAS induced transformation
- 11) References are incorrectly placed and many of them are missing (e.g., 15 and 39)

Referee #3:

SUMMARY:

This manuscript is scientifically and methodologically very strong. The findings add a considerable amount of new knowledge to what is known about pancreatic carcinogenesis and the underlying genes and transcriptional networks that are important in nudging the normal pancreas (in a state of homeostasis) towards de-differentiation and oncogenesis. The focus of the manuscript is HNF1A, a transcription factor known to be important for the endocrine pancreas and recently implicated as being a tumor suppressor for the exocrine pancreas through analyses in vitro. The authors show that HNF1A loss synergizes in a dramatic manner with KRasG12D in causing pancreatic cancer with sarcomatoid features in mice as young as 8-10 weeks of age. They also show that KDM6A loss acts in much the same manner, and that HNF1A recruits KDM6A to regulatory elements of genes important for suppressing the expression of EMT promoting and oncogenic promoting genes. Therefore, in summary, the authors show convincingly that HNF1A is indeed a tumor suppressor for the pancreas in vivo and link this transcription factor to KDM6A, a gene known to be mutated in a considerable amount of human pancreatic cancers.

MAJOR CONCERNS

1. Gene names throughout the manuscript need to be changed to UniProt, HUGO and MGI nomenclature. For KDM6A the official protein name (UniProt) is Lysine-specific demethylase 6A and the official gene name (HUGO and MGI) is KDM6A. Note that UTX is a previous acronym for this gene and should not be used (apart from perhaps once stating that it is a previous gene name for KDM6A)

2. GWAS results for pancreatic cancer. On page 3 the authors state that "HNF1A locus contains one of the strongest PDAC genetic association signals.....". This is not correct. In fact, the locus is suggestive as the P value does not reach the Bonferroni corrected P-value of 5×10^{-8} that is the accepted threshold for robust GWAS findings. In the latest pancreas cancer GWAS by Klein et.al: Genome-wide meta-analysis identifies five new susceptibility loci for pancreatic cancer (PMID: 29422604), this locus is listed with the following results:

Chr12q24: $P=3.5 \times 10^{-7}$, OR=1.11

This means that the chr12q24 locus is GWAS suggestive but not GWAS significant. In fact, the best locus for that paper has a P value (see table 2 of same paper) of $P=10^{-15}$ which is much more significant.

In a GWAS pathway analysis (Genome Wide Pleiotropy Scan identifies HNF1A, PMID: 21498636), the authors indicate that the strongest association they observed was for SNP rs7310409 ($P = 3 \times 10^{-5}$). However, the strongest association in a pathway based GWAS analysis (here called pleiotropy scan) is not the same as the strongest association in a GWAS.

This needs to be stated in the paper - i.e. that this locus has a suggestive but not genome wide significant GWAS signal.

MINOR CONCERNS

3. Page 6: "Furthermore, the analysis of genotypes from the Australian ICGC study revealed putative loss of function UTC mutations"

Please clarify or change the word "genotypes" to mean "somatic genotypes" or "somatic mutations". Genotypes per se can mean many things (germline and somatic) and the wording is not very clear in this sentence.

4. Some P values are quite exact (e.g. $P=2.9 \times 10^{-6}$ and $P=0.0064$) whereas others are approximate or less than a certain number (e.g. $P < 10^{-4}$). Please show all P values as exact numbers.

1st Revision - authors' response

5th Dec 2019

POINT BY POINT RESPONSE

We are grateful to all three reviewers for the careful evaluation and constructive comments, which we have addressed as detailed below. Please note that supplementary figures have now been renamed as follows:

Supplementary Figure 1 -> Appendix Figure S1

Supplementary Figure 2 -> Figure EV 1

Supplementary Figure 3 -> Figure EV 2

Supplementary Figure 4 -> Appendix Figure S2

Supplementary Figure 5 -> Figure EV 3

Supplementary Figure 6 -> Figure EV 4

Supplementary Figure 7 -> Figure EV 5

Referee #1:

Overall, this is a very interesting study that sheds light on how epigenetic modifiers, such as UTX, work together with lineage-specific transcription factors, such as HNF1A, in the context of pancreatic differentiation and cancer. All the genetics, transcriptome and biochemical data are very convincing and support well the proposed model of a direct positive regulatory control of UTX through HNF1A on pancreatic differentiation. By contrast, the evidence for an indirect negative control on EMT and oncogenic pathways is tenuous - this could be expanded or better discussed by the authors.

We thank the reviewer for the praiseful comments and useful suggestion.

To address the suggestion, we list numerous examples of HNF1A/UTX direct target genes that show downregulation in the KO models and are known negative regulators of EMT, including *FoxA3* (PMID:29155818), *Deptor* (PMID:31685947), and negative regulators of MAPK signaling such as *Gstp1* (PMID: 11408560 and PMID: 16023107) and *DEP-1/Ptprj* (PMID: 19494114), which are relevant because ERK signaling is known to be essential for EMT (PMID: 24556840). We mention these example genes in the results section (line 311 and 317).

In addition to the lack of enrichment of HNF1A/UTX binding, we provide more examples of EMT-associated genes that show upregulation in HNF1A KO and have no evidence of HNF1A binding within 300 Kb (**Figure 2J**).

In the discussion, we emphasize the arguments that support our conclusion that effects on EMT and oncogenic pathways are indirect as follows (line 372-378):

Our integrated analysis of genomic binding sites indicated that the positive regulatory effects of KDM6A (and HNF1A) on acinar differentiation genes are largely mediated through direct mechanisms. By contrast, genes associated with EMT and oncogenic pathways that were upregulated in KO models were not frequently bound by HNF1A and KDM6A, while both proteins bound and activated known negative regulators of EMT. These findings strongly suggest that the suppression of oncogenic and EMT transcriptional programs is predominantly indirect.

Additional comments:

- The authors should further explain or discuss why a non-classical PDAC sarcomatoid architecture is associated with HNF1A/UTX-deficient molecular signature. Is it because of the ECM remodelling or the EMT genes dysregulation?

Sarcomatoid carcinomas are thought to reflect profound EMT, and molecular studies (including the current study) point to upregulation of EMT markers, as well as ECM genes, which in turn have been tightly linked to EMT. The relative role of different EMT vs. specifically genes involved in ECM genes not been tested, so it is difficult for us to speculate on the relative contribution of different mechanisms.

We have now, however, briefly emphasized extracellular matrix remodeling together with EMT in the discussion (line 374-375).

- Looking at the histology in Fig.1A and Fig. S5i, I wonder if there is any consequence in the pancreatic tissue-architecture and lobule organization in the HNF1A- and UTX-single KO animals. If this can be excluded, then the micrographs shown in Fig.1a and Fig. S5i are not representative and should be replaced.

The histology in young mice is normal in both KO models. **Figure 1A** and **Figure EV 3I**, which are mentioned by this reviewer, are immunofluorescence stainings which are not ideal to assess morphology. We prefer to highlight the HE images in **Appendix Figure S1(D)**. To we have modified the text to make a more unequivocal statement in this regard (line 57-61):

As expected from previous studies of Hnf1a germ-line null mutants, this did not produce gross defects in pancreas organogenesis or tissue architecture

The acinar-specific (in addition to the pancreas-specific) HNF1A KO also showed normal morphology, as shown in HE stainings for HNF1A KO in **Figure EV 1B and 1C**. We note that **Figure 1A** and **Figure EV 1C** have been updated in the new version to present better quality images.

We do wish to mention, however, that we have noticed that by 8 weeks of age UTX-KO do show some signs of acinar cell attrition and pancreatic lobe atrophy with fat replacement, which we now know progress with time. This was already shown in the zoomed image in **Figure EV 3H'**, and we are now mentioning it in the main text (line 228). The conclusion remains unchanged, in that neither HNF1A or UTX are required for organogenesis or organ architecture at stages that precede the formation of tumors that we observe in the KRAS mutants.

- Is it known where UTX is expressed in normal pancreatic tissue?

We now provide images of UTX expression in normal pancreas, showing it is expressed in acinar, duct and endocrine cells (**Appendix Figure S2C**).

Referee #2:

This is an elegantly done manuscript by Kalisz and colleagues supporting a role for the functional interaction between UTX and HNF1A in the development of Kras-induced pancreatic tumor initiation. The investigational team used a set of state-of-the-art animal models and genome wide studies to evaluate the transcriptional programs used by this complex in the regulation of gene expression during pancreatic transformation. The data and writing are clearly presented, however, additional results are needed to fully determine the impact of the UTX-HNF1A complex in the regulation of pancreatic carcinogenesis. Specifically, the nature of the interaction, the role in human models as well as lack of key controls.

We thank this reviewer for these complimentary comments.

Specific comments:

1) The effect HNF1A loss on the expression UTX should be evaluated as the magnitude of the drop in UTX enrichment is difficult to account based on the number of HNF1A targets. The differences could be explained by a decreased expression of UTX and not lack of recruitment to target genes.

Agreed. We have incorporated this important control, which is now shown in **Figure 7B**.

2) The validation of key findings (cell growth, target gene regulation and interaction - IP and CHIP) should be done in pancreatic human cells (KRAS-transformed and non-transformed). This will enhance the disease significance of the study.

The focus of this study, as discussed in the original submission, is on the function of HNF1A and UTX in acinar cells, which are thought to be the cells of origin for common forms of PDAC. Our analysis of non-transformed mouse KO pancreas shows that HNF1A regulates a cancer-relevant program in acinar cells. The validity of these molecular findings for human cells should ideally be tested in non-transformed human acinar cells. To the best of our knowledge human acinar cell lines do not exist while primary cultures of human acinar cells have been challenging to establish over the years. Once in culture they rapidly change their phenotype, downregulate acinar markers and dedifferentiate to progenitor like cells or transdifferentiate to ductal like cells with EMT features (PMID: 21703267; PMID: 1373187; PMID: 12870182; PMID: 25003220; PMID: 30858455). We can thus unfortunately not address this point by perturbing HNF1A in human acinar cells.

In our previous manuscript, we did show that mouse KO molecular signatures can be observed in a subset of human tumors. We now additionally show that mouse KO expression phenotypes correlate with HNF1A mRNA levels in human pancreas RNA-seq samples. This analysis was carried out with expression data from 328 non-diseased human pancreas from the Genotype-Tissue Expression (GTEx)

project. We compared samples with lowest vs. highest *HNF1A* mRNA deciles, and show that human orthologs of up- and downregulated genes in *Hnf1a* and *Kdm6a* KO varied as a function of HNF1A mRNA levels. This result has been incorporated in **Figure 3A**, and line 117-131.

3) ChIP-seq using genome browser track is not ideal, at least some key genes and chromatin landscaping mapping should be validated by ChIP-PCR in tissue or cell lines derived from the WT and KO models. Also, the levels of of H3K4me should be evaluated in both WT and KO cells.

We have performed qPCRs as suggested.

Figure EV 4 now shows qPCR for HNF1A and UTX binding at co-bound regions, as well as mRNA levels in *Hnf1a* KO pancreas in 4 loci. **Figure EV 3T,U** shows mRNA levels in both KO in two co-bound loci.

We have also performed ChIP-qPCR of H3K27me3, H3K27ac and H3K4me1 in WT and UTX-KO pancreatic chromatin in representative bound regions, and RT-PCR for those genes. (**Figure 5H,I** and **Figure EV 3Q,R**; main text line 280-287).

All validations confirmed NGS-based findings.

4) The levels of Kras activation should be evaluated in both WT and KO models by examining the levels of pERK.

We evaluated levels of phospho-ERK in our models by immunoblots, and found increased levels of phospho-p42 (phospho-ERK2) in HNF1A and UTX mutants (**Figure EV 1F** and **Figure EV 3K**).

Given that this study has focused the functional analysis on non-transformed mouse models we have performed this analysis in these mutant models. In fact, we do not necessarily expect a further increase in transformed cells, which already show high levels.

5) H/E of the Pdx1-HNF1A KO should be shown (high and low magnifications) and low magnification of the UTX KO, the IF images are insufficient to evaluate the impact of the KOs on the histomorphology of the pancreas. The positive signal (Sup Fig 1C) should be explained, it seems that HNF1A KO still has some acinar signal?

We have now included representative H-E pictures with high and low magnifications of *Pdx1Cre-Hnf1a* KO in **Appendix Figure S1D** and low magnification of *Pdx1Cre-UTX* KO pancreas in **Figure EV 3F**. Both mutants show normal morphology up to 8 weeks. Beyond that, acinar cell attrition is observed, with fat cell replacement, as shown in **Figure EV 3H'**.

The positive HNF1A signal in **Appendix Figure S1C**, is likely due to lack of Cre mediated recombination in a very small number of cells. Incomplete deletion is commonly observed in the pancreas when using transgenic Cre drivers. The *Pdx1Cre* line selected for this study is regarded as the most efficient *Pdx1Cre* line available, and excised the vast majority of cells (as shown in IF and immunoblot studies in **Figure EV 3A** and **Appendix Figure S2D**).

6) It will be ideal (but not critical) that the TMAs are co-stained for UTX and the UTX mutational status should be included.

The contingency tables in **Figure EV 2F** showing that UTX levels correlate with tumor grade are based on co-staining for UTX. We do not have the UTX mutational status for the samples analyzed in TMAs as they were not DNA sequenced.

7) Validation of HNF1A and UTX targets by CHIP-PCR and qPCR is a must. This should be done on tissue or cell lines derived from the mouse models. It is important to define the co-occupancy of both molecules in a selected group of target genes. Also, it will be important to determine if the recruitment to target genes is induced by mutant KRAS?

As also explained above, we have now performed qPCRs confirmations of main findings. **Figure EV 4** now shows qPCR for HNF1A and UTX binding at co-bound regions, as well as mRNA levels in *Hnf1a* KO pancreas in 4 loci. All validations confirmed NGS-based findings.

We have not analysed if the recruitment is induced further by mutant KRAS, because we already observe that UTX recruitment is nearly completely lost in HNF1A KO, without KRAS. We do not expect further changes in transformed PDAC cell lines.

8) The requirement for UTX for the HNF1A-dependent transcription should be examined using RNAi and overexpression tools in cell lines models. So far, the authors have shown associations to support this event, not a cause-effect.

Our studies have demonstrated causality based on transcriptional changes that occur as a consequence of directed conditional mutations, which is more than an association. Because human acinar lines do not exist, we have now created UTX-KO clones in a mouse acinar cell line by CRISPR-Cas9. We confirmed by qPCR that HNF1A-bound genes are down-regulated in the UTX-KO clones, while HNF1A binding to those genes was unaffected by the loss of UTX expression, as observed in the KO studies. These findings confirm the functional dependence on UTX for transcriptional regulation by HNF1A in a separate in vitro cell model. The results have been added to a new **Figure EV 5B-F** (see also page 343-346).

9) The histological and transcriptional differences with Cancer Cell article from Tzatzos' group should be discussed.

Thank you for this comment. We now show that gene expression changes in pancreas from *Hnf1a* pKO and *Kdm6a* pKO are markedly deregulated in *Kras*^{G12D}/*Kdm6a* KO tumors reported in Andricovich et al, Cancer Cell. Furthermore, the functional annotations that were most enriched in deregulated genes in tumors (Supplementary Figure 3 in Andricovich et al, Cancer Cell) showed significant enrichment in *Hnf1a* pKO and *Kdm6a* pKO deregulated genes. Understandably, some annotations, including “MycMax” and “pancreatic cancer” were not enriched in our non-tumoral samples. We now provide this comparison in **Figure EV 5G,H**, and describe the data briefly in line 322-327 as follows:

We further found that transcriptional changes in HNF1A- and KDM6A-deficient pancreas were also recapitulated in Kdm6a-mutant Kras^{G12D} tumors (Andricovich) (Fig EV 5G,H). This provided further evidence that HNF1A function is not only relevant to Kdm6a-regulated programs in the non-tumoral pancreas, but also to Kdm6a-deficient cancer.

Concerning the histology of pancreatic tumors in the Andricovich et al, Cancer Cell study, the images are largely consistent with the tumors from our study. One important distinction, however, is that Andricovich et al interpret that *Kdm6a* KO tumors are squamous-like whereas we have not identified specific morphological features that define squamous differentiation. We should note that we also did not identify unequivocal signs of squamous/adenosquamous carcinoma in the representative image provided in the Andricovich manuscript (Figure 2B). We have now noted that we do not find signs of adenosquamous carcinoma (line 203-205).

These findings were consistent with recently reported observations in 6 week-old mice, with the exception that we have not observed signs of squamous differentiation (Andricovich et al., 2018).

It is perhaps worth emphasizing that “adenosquamous carcinoma” morphology is a very rare form of PDAC that is not necessarily apparent in the relatively large subset of human tumors that exhibit a “squamous-like” molecular signature.

10) The authors should evaluate (in an in vitro system) the impact of some of the genes commonly regulated the transcriptional program regulated by HNF1A-UTX in KRAS induced transformation

Our studies point to an extremely broad HNF1A/UTX-dependent transcriptional program (see **Figure 2C,D, Figure 5C,D, Table EV2, 4 and 5**). This includes numerous transcription factors and metabolic genes that are directly bound by HNF1A/UTX. Some of these, in turn, are well known negative regulators of EMT, ECM remodelling and/or MAPK signalling (see examples cited in the response to the first query from reviewer #1). As pointed out in the manuscript, the oncogenic role of several transcriptional changes in our KO models, at the pathway or gene level, has already been established. On the other hand, given the large number of genes affected, it is not necessarily true that the consequences of HNF1A or UTX deficiencies can be ultimately ascribed to a single gene. Many deregulated genes are likely to play a role in aggregate. We therefore do not necessarily expect that perturbation of single target genes will result in increased KRAS-induced transformation. We do not believe that a screen to identify additional individual target genes that affect KRAS-induced transformation on their own would modify the validity of our conclusions. We thus respectfully refrain from carrying out these experiments in the context of this study, and hope that this reviewer agrees with this argumentation.

11) References are incorrectly placed and many of them are missing (e.g., 15 and 39)

We thank the referee for noticing these errors which we now have corrected. We have also gone through all references to ensure they are accurate.

Referee #3:

SUMMARY:

This manuscript is scientifically and methodologically very strong. The findings add a considerable amount of new knowledge to what is known about pancreatic carcinogenesis and the underlying genes and transcriptional networks that are important in nudging the normal pancreas (in a state of homeostasis) towards de-differentiation and oncogenesis.

The focus of the manuscript is HNF1A, a transcription factor known to be important for the endocrine pancreas and recently implicated as being a tumor suppressor for the exocrine pancreas though analyses in vitro. The authors show that HNF1A loss synergizes in a dramatic manner with KRasG12D in causing pancreatic cancer with sarcomatoid features in mice as young as 8-10 weeks of age. They also show that KDM6A loss acts in much the same manner, and that HNF1A recruits KDM6A to regulatory elements of genes important for suppressing the expression of EMT promoting and oncogenic promoting genes. Therefore, in summary, the authors show convincingly that HNF1A is indeed a tumor suppressor for the pancreas in vivo and link this transcription factor to KDM6A, a gene known to be mutated in a considerable amount of human pancreatic cancers.

Thank you for the flattering note, we have addressed concerns as shown below.

MAJOR CONCERNS

1. Gene names throughout the manuscript need to be changed to UniProt, HUGO and MGI nomenclature. For KDM6A the official protein name (UniProt) is Lysine-specific demethylase 6A and the official gene name (HUGO and MGI) is KDM6A. Note that UTX is a previous acronym for this gene and should not be used (apart from perhaps once stating that it is a previous gene name for KDM6A)

We have now corrected UTX to KDM6A throughout the manuscript adhering to the suggested nomenclatures.

2. GWAS results for pancreatic cancer. On page 3 the authors state that "HNF1A locus contains one of the strongest PDAC genetic association signals.....". This is not correct. In fact, the locus is suggestive as the P value does not reach the Bonferroni corrected P-value of 5×10^{-8} that is the accepted threshold for robust GWAS findings. In the latest pancreas cancer GWAS by Klein et.al: Genome-wide meta-analysis identifies five new susceptibility loci for pancreatic cancer (PMID: 29422604), this locus is listed with the following results: Chr12q24: $P=3.5 \times 10^{-7}$, $OR=1.11$. This means that the chr12q24 locus is GWAS suggestive but not GWAS significant. In fact, the best locus for that paper has a P value (see table 2 of same paper) of $P=10^{-15}$ which is much more significant. In a GWAS pathway analysis (Genome Wide Pleiotropy Scan identifies HNF1A, PMID: 21498636), the authors indicate that the strongest association they observed was for SNP rs7310409 ($P = 3 \times 10^{-5}$). However, the strongest association in a pathway based GWAS analysis (here called pleiotropy scan) is not the same as the strongest association in a GWAS. This needs to be stated in the paper - i.e. that this locus has a suggestive but not genome wide significant GWAS signal.

We appreciate that this was an overstatement, and have modified this, we now reworded this as follows (line 35-36):

Furthermore, GWA studies suggest that genetic variants in the HNF1A locus predispose to PDAC (Pierce & Ahsan, 2011, Klein et al., 2018).

MINOR CONCERNS

3. Page 6: "Furthermore, the analysis of genotypes from the Australian ICGC study revealed putative loss of function UTC mutations"

Please clarify or change the word "genotypes" to mean "somatic genotypes" or "somatic mutations". Genotypes per se can mean many things (germline and somatic) and the wording is not very clear in this sentence.

We have now reworded this sentence to (line 179-181):

Furthermore, analysis of the Australian ICGC-PACA data revealed putative loss of function KDM6A mutations in 19% of tumors showing HNF1A LoF phenotypes, compared with 2% of all other tumors

4. Some P values are quite exact (e.g. $P=2.9 \times 10^{-6}$ and $P=0.0064$) whereas others are approximate or less than a certain number (e.g. $P < 10^{-4}$). Please show all P values as exact numbers.

We are now showing exact P values throughout the manuscript.

Thank you in advance for assessing this re-submission, with best wishes

2nd Editorial Decision

8th Jan 2020

Thank you for submitting your revised manuscript for consideration by The EMBO Journal. My apologies for getting back to you with delay at this time of the year due to protracted referee input. Your amended study was sent back to two of the referees for re-evaluation, and we have received comments from both of them, which I enclose below.

As you will see the referee finds that their concerns have been sufficiently addressed and they are now broadly in favour of publication.

Thus, we are pleased to inform you that your manuscript has been accepted in principle for publication in The EMBO Journal, pending some minor issues related to formatting and data representation as listed below, which need to be adjusted at re-submission.

REFeree REPORTS:

Referee #1:

In the revised manuscript the authors addressed all concerns requested by this reviewer and also added some additional experiments, which clarified and reinforced their observations.

The present version of the manuscript is now suitable for publication in The EMBO Journal.

Referee #2:

The authors have been responsive to reviewers' criticisms. Now, the major conclusions of the study are supported for previous (original submission) and new data set. Also, the addition of key sections in the write-up addressed the limitations of the study. This is very valuable as it sets up the foundation for future studies.

2nd Revision - authors' response

2nd Feb 2020

The authors performed the requested editorial changes.

Thank you for submitting the revised version of your manuscript. I have now evaluated your amended manuscript and concluded that the remaining minor concerns have been sufficiently addressed.

Thus, I am pleased to inform you that your manuscript has been accepted for publication in the EMBO Journal.

Corresponding Author Name: Jorge Ferrer

Journal Submitted to: The EMBO Journal

Manuscript Number: EMBOJ-2019-102808R